# Phenylalanine impairs insulin signaling and inhibits glucose uptake through modification of IRβ

Qian Zhou[1,2,3,7], Wan-Wan Sun[3,7], Jia-Cong Chen[1,2], Hui-Lu Zhang[3], Jie Liu[1,3], Yan Lin[1,2], Peng-Cheng Lin[4], Bai-Xing Wu[5], Yan-Peng An[1], Lin Huang[1], Wen-Xing Sun[6], Xin-Wen Zhou[1], Yi-Ming Li[3], Yi-Yuan Yuan ●[1,2], Jian-Yuan Zhao ●[1,2], Wei Xu ●[1,2,3] ✉ & Shi-Min Zhao ●[1,2,4] ✉

Whether amino acids act on cellular insulin signaling remains unclear, given that increased circulating amino acid levels are associated with the onset of type 2 diabetes (T2D). Here, we report that phenylalanine modifies insulin receptor beta (IRβ) and inactivates insulin signaling and glucose uptake. Mice fed phenylalanine-rich chow or phenylalanine-producing aspartame or over-expressing human phenylalanyl-tRNA synthetase (hFARS) develop insulin resistance and T2D symptoms. Mechanistically, FARS phenylalanylate lysine 1057/1079 of IRβ (F-K1057/1079), inactivating IRβ and preventing insulin from promoting glucose uptake by cells. SIRT1 reverse F-K1057/1079 and counteract the insulin-inactivating effects of hFARS and phenylalanine. F-K1057/1079 and SIRT1 levels in white blood cells from T2D patients are positively and negatively correlated with T2D onset, respectively. Blocking F-K1057/1079 with phenylalaninol sensitizes insulin signaling and relieves T2D symptoms in *hFARS*-transgenic and db/db mice. These findings shed light on the activation of insulin signaling and T2D progression through inhibition of phenylalanylation.

To facilitate glucose uptake, insulin is secreted by beta cells in islets upon glucose exposure, and insulin signaling is activated in other cells, such as muscle cells, to promote glucose uptake after glucose consumption[1]. Insulin signaling is initiated by the binding of insulin to the α subunit of the insulin receptor (IRα), which activates the tyrosine kinase activity of β subunit of the insulin receptor (IRβ) by phosphorylating tyrosine residues in the cytoplasmic domain of IRβ[2].

Insulin signaling proceeds through the recruitment of the IR substrate (IRS) family proteins[3] via phosphorylated IRβ and subsequent activation of insulin signaling components, namely, PI3K, Akt, and AS160, to promote GLUT4-mediated glucose uptake[4]. Impairments in any step of the insulin cascade cause defects in glucose uptake and result in diabetes[5]. For example, defects in insulin secretion cause maturity-onset diabetes in the young (MODY). Impairments in the components

[1]Obstetrics & Gynecology Hospital of Fudan University, State Key Laboratory of Genetic Engineering, School of Life Sciences and Institutes of Biomedical Sciences, Fudan University, 200438 Shanghai, P.R. China. [2]NHC Key Lab of Reproduction Regulation (Shanghai Institute of Planned Parenthood Research), Institute of Metabolism and Integrative Biology, Shanghai Key Laboratory of Metabolic Remodeling, and Children's Hospital of Fudan University, 200438 Shanghai, P.R. China. [3]Endocrinology department, Huashan Hospital, 5th affiliated Hospital, Fudan University Shanghai Cancer Center, Fudan University, 200438 Shanghai, P.R. China. [4]Key Laboratory for Tibet Plateau Phytochemistry of Qinghai Province, College of Pharmacy, Qinghai University for Nationalities, 810007 Xining, P. R. China. [5]Guangdong Provincial Key Laboratory of Malignant Tumor Epigenetics and Gene Regulation, Guangdong-Hong Kong Joint Laboratory for RNA Medicine, RNA Biomedical Institute, Medical Research Center, Sun Yat-Sen Memorial Hospital, Sun Yat-Sen University, 510120 Guangzhou, China. [6]Department of Nutrition and Food Hygiene, School of Public Health, Nantong University, 226019 Nantong, China. [7]These authors contributed equally: Qian Zhou, Wan-Wan Sun. ✉e-mail: xuwei_0706@fudan.edu.cn; zhaosm@fudan.edu.cn

of the insulin signaling cascade, such as competition between isoforms of IR[6] that disrupt insulin signaling generation or deletion of *AKT2* resulting in disruption of insulin signaling transmission[7], cause reduced insulin sensitivity and type 2 diabetes (T2D).

Glycerol from triglyceride and amino acids are two other major energy sources that can be converted to glucose through gluconeogenesis when the glucose supply is low. Physiologically, when cells sense high levels of intracellular fatty acids or amino acids, they may suppress glucose uptake to avoid an oversupply of energy nutrients and a waste of resources, because glucose uptake consumes energy and other cellular resources. Fatty acids have been shown to suppress insulin signaling and glucose uptake through various mechanisms. Free fatty acids inhibit insulin signaling by inhibiting AKT kinase activity[8]; unsaturated C18 fatty acid elaidate suppresses the insulin-induced fusion of GLUT4 storage vesicles in the plasma membrane[9], and palmitic acid induces sustained JNK activation and insulin resistance in primary mouse hepatocytes and pancreatic β-cells[10].

In contrast to fatty acids, the roles of amino acids in regulating insulin signaling and glucose uptake are unclear, although increased blood levels of aromatic amino acids, branched-chain amino acids, and other amino acids have been repeatedly found in T2D patients[11–13], suggesting that deregulated amino acid levels either cause or are a consequence of T2D. Moreover, habitually high-protein intake, which is often employed to decrease body weight, increases amino acid levels and is associated with insulin resistance and an increased risk of developing T2D[14,15], supporting the hypothesis that upregulated amino acids may be an underlying cause of the disease. Mechanistically, amino acid infusion or excessive protein ingestion reduces insulin sensitivity independent of mTOR signaling activation[16], suggesting that amino acids may not impair insulin-mediated glucose disposal mainly by activating anabolism, such as protein synthesis, although essential amino acids stimulate skeletal muscle mitochondrial protein synthesis, which accounts for a minor fraction of total cellular protein synthesis, and decreases insulin sensitivity in healthy humans[17]. Moreover, amino acids may induce insulin resistance in human skeletal muscle by inhibiting glucose transport[18]. These results link amino acid oversupply to insulin resistance, inhibition of glucose uptake, and progression to T2D through mostly unknown mechanisms.

Fatty acids can modify proteins as substrates to alter cell signaling[19]. Amino acids are newly identified modifiers of proteins. Aminoacyl tRNA synthetases (ARSs) are aminoacyl transferases that catalyze the lysine aminoacylation of its cognate amino acid to proteins to regulate a number of physiological and pathological processes, including mTORC1 signaling[20–22], apoptosis[21], and neuron loss[20]. In the current study, we provide evidence that phenylalanine, one of the amino acids exhibiting significantly increased levels in the sera of T2D patients[11–13], induces insulin resistance and T2D symptoms by modifying IRβ.

## Results

### Phenylalanine-rich and aspartame-containing chow induced T2D symptoms in mice

To elucidate whether phenylalanine elevation is a cause or consequence of T2D, we challenged C57BL/6J mice (C57 mice) with a phenylalanine-rich chow. Male C57 mice were fed chow containing 1% (10 g/kg) phenylalanine for 12 weeks, during which food intake and body weight were monitored over time, and other tests were performed at the 12th week. Although mouse serum phenylalanine levels increased by 75% to reach approximately 140 μM in phenylalanine-rich chow-fed mice (Fig. 1a), simulating 20-25% phenylalanine elevation in T2D human sera samples[23,24], there were negligible differences in caloric intake (Supplementary Fig. 1a) or body weights (Supplementary Fig. 1b) between the normal chow- and phenylalanine-rich chow-fed mice. However, fasting blood glucose levels were elevated (Fig. 1b), glucose tolerance was reduced as assayed by the glucose tolerance test

(GTT) (Fig. 1c, Supplementary Fig. 1c), insulin tolerance was reduced as assayed by the insulin tolerance test (ITT) (Fig. 1d), blood insulin levels were increased (Fig. 1e), and the homeostasis model assessment of insulin resistance (HOMA-IR), indicators of insulin resistance, and T2D[25] (Fig. 1f) were induced by phenylalanine-rich chow feeding. To mimic a more physiologically relevant phenylalanine exposure, we used 0.1% phenylalanine chow, which increased blood phenylalanine by 22.9% (Supplementary Fig. 1d), and this manipulation resulted in insulin resistance and T2D phenotypes after 6 months of feeding (Supplementary Fig. 1e–k), slower than the 1% phenylalanine chow treatments. These results suggest that phenylalanine promotes T2D onset in mice in a dose-dependent manner. Further confirming this notion, a chow containing 1% aspartame, which elevated mouse serum phenylalanine levels by 36% to reach approximately 120 μM (Supplementary Fig. 1l) and increased body weight (Supplementary Fig. 1m) and increased food intake (Supplementary Fig. 1n), induced elevated blood glucose (Supplementary Fig. 1o) and blood insulin (Supplementary Fig. 1p) levels and reduced glucose tolerance (Supplementary Fig. 1q, r), insulin tolerance (Supplementary Fig. 1s), and elavated HOMA-IR value (Supplementary Fig. 1l) that phenocopied those of phenylalanine-rich chow after 12 weeks of feeding (Figs. 1c–f).

### Intracellular phenylalanine blocked insulin signaling in cultured cells

To elucidate the underlying T2D-promoting mechanism of phenylalanine, we tested the effects of phenylalanine supplementation on cultured cells. Methyl-phenylalanine (Me-Phe) supplementation of 3T3-L1 murine adipocytes and L6 rat skeletal myoblasts in 2 mM culture media, which can be converted to phenylalanine by intracellular esterase[26], successfully increased intracellular phenylalanine levels approximately six-fold (Supplementary Fig. 2a, 2b). Moreover, Me-Phe supplementation severely prevented insulin from stimulating glucose uptake in these cells (Fig. 1g, Supplementary Fig. 2c), as measured by 2-deoxy-D-glucose (2-DG) uptake assays, suggesting that phenylalanine may blunt insulin signaling. This notion was confirmed by Me-Phe supplementation, which prevented insulin from increasing the phosphorylation levels of IR tyrosine 1150/1151, insulin receptor substrate 1 (IRS1) tyrosine, AKT threonine 308 and serine 473, and AS160 threonine 642, indicators of insulin signaling activation, in 3T3-L1 adipocytes (Figs. 1h, 1i) and L6 cells (Supplementary Fig. 2d, 2e) in a time- and dose-dependent manner before the self-decay of insulin signaling (Fig. 1h, Supplementary Fig. 2d). Furthermore, membrane enrichment of GLUT4, a consequence of insulin signaling activation[27], was prevented by Me-Phe treatment in 3T3-L1 adipocytes (Fig. 1j).

Insulin signaling is initiated in the extracellular compartment and is transmitted intracellularly. Me-Phe treatment blocked insulin signaling, indicating that phenylalanine acts intracellularly to block insulin signaling. This conclusion was further supported by the fact that, in 3T3-L1 cells, small interfering RNA (siRNA)-mediated silencing of phenylalanine dehydrogenase (*PAH*), the first phenylalanine catabolizing enzyme, to increase intracellular phenylalanine levels (Supplementary Fig. 2f), effectively desensitized insulin to activate insulin signaling in 3T3-L1 cells (Fig. 1k), but did not noticeably change the protein levels of the insulin signaling pathway (Supplementary Fig. 2g). These results suggest that intracellular phenylalanine acts on one or more components of and inactivates insulin signaling.

### FARS overexpression phenocopied phenylalanine treatment efficacy

To examine whether phenylalanine inhibits insulin signaling by augmenting lysine phenylalanylation (phenylalanylation hereafter), a reported posttranslational modification that is catalyzed by phenylalanyl-tRNA synthetase (FARS) and employing phenylalanine as a substrate[21], we checked whether cytosolic FARS, mitochondrial phenylalanyl-tRNA synthetase 2 (FARS2), or both are involved in

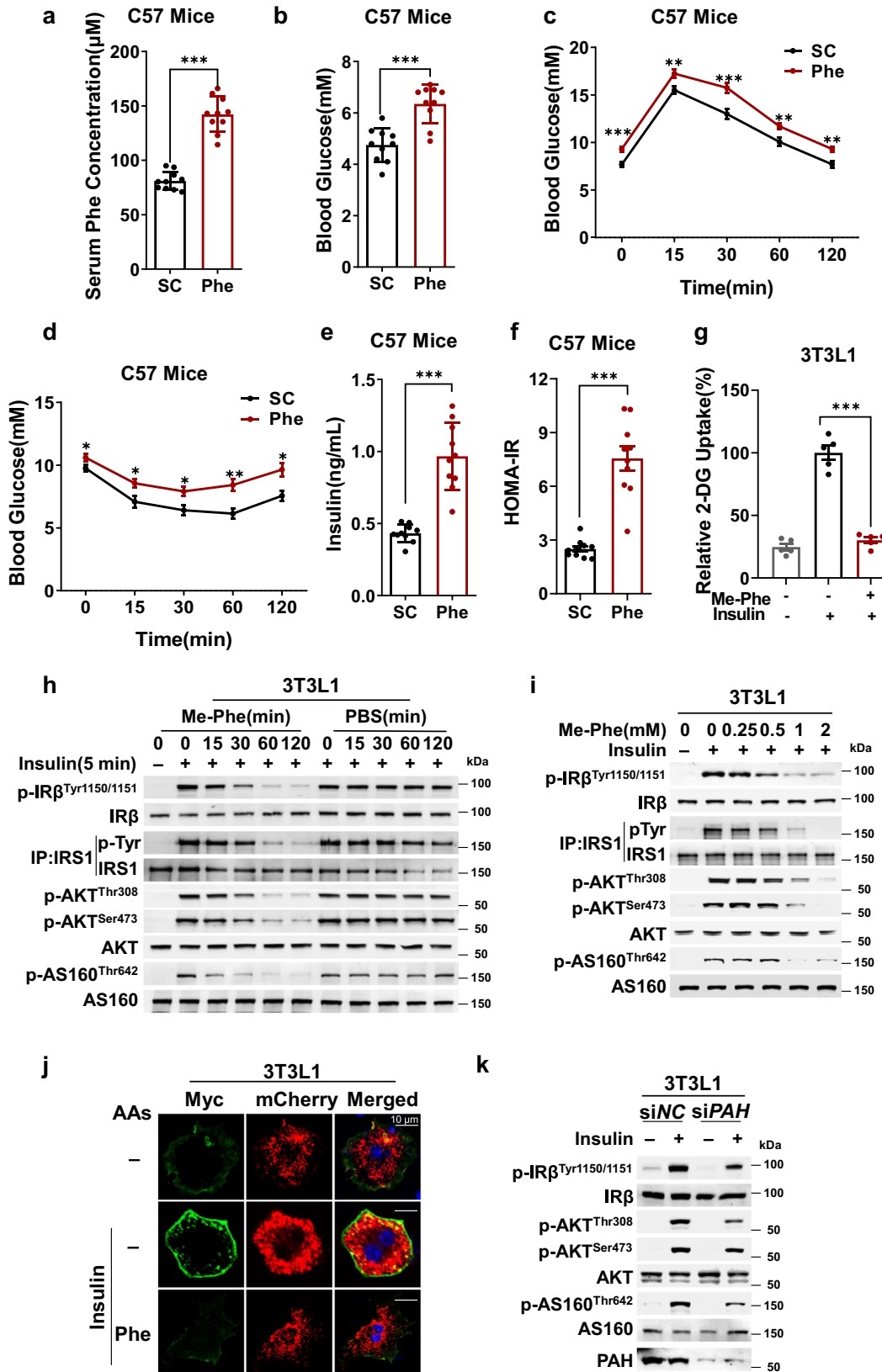

insulin signaling regulation. Ectopically co-expressing both the α and β subunits of FARS (FARSA and FARSB), but not FARS2 (Supplementary Fig. 3a), blunted insulin signaling (Fig. 2a). Insulin signaling inactivation by FARS was substantiated by the insulin-stimulated 2-DG uptake ability of 3T3-L1 cells, which was partially abrogated by FARS overexpression (Fig. 2b) and further by silencing catalytic *FARSA*[28] but not

*FARS2* (Supplementary Fig. 3b) with siRNA, sensitizing insulin to activate insulin signaling (Fig. 2c), and promoting 2-DG uptake (Fig. 2d) in 3T3-L1 cells. Moreover, knockout of *FARSA* in HepG2 cells or knockdown *of FARSA* in 3T3-L1 cells sensitized insulin signaling (Fig. 2e) and insulin-promoted 2-DG uptake (Fig. 2f), abrogating Me-Phe to inhibit insulin signaling and 2-DG uptake (Figs. 2e, 2f). These results suggest

**Fig. 1 | Phenylalanine induces symptoms of type 2 diabetes (T2D) in mice and cells. a–f** Phenylalanine-supplemented chow induces T2D symptoms in mice. Male C57 mice (n = 10) were fed standard chow (SC) or phenylalanine-supplemented chow (Phe). Relative blood Phe (**a**), blood glucose (**b**), glucose tolerance (**c**), insulin tolerance (**d**), and blood insulin levels (**e**) and HOMA-IR values (**f**) were measured in fasted mice after 12 weeks of feeding. **g** Me-Phe treatment impairs insulin-stimulated 2-DG uptake. 2-DG Uptake by insulin-stimulated 3T3-L1 differentiated adipocytes were detected in the absence or presence of Me-Phe in the culture media (n = 5). h-i, Me-Phe treatment impairs insulin signaling. Time- (**h**) and dose-dependent (**i**) effects of Me-Phe on the phosphorylation of IR, IRS1, AKT, and AS160 in 3T3-L1 adipocytes were detected. Insulin signaling was activated before treatment to detect the inhibition of insulin signaling. **j** Me-Phe treatment decreases GLUT4 membrane translocation. Membrane GLUT4 levels were detected by immunofluorescence in 3T3-L1 adipocytes and insulin-stimulated 3T3-L1 adipocytes in the absence or presence of Me-Phe in the culture media. **k** PAH knockdown impairs insulin signaling. The effects of insulin on the phosphorylation of IR, AKT, and AS160 were detected in 3T3-L1 adipocytes and in 3T3-L1 adipocytes after *PAH* knockdown. Student's *t* tests (unpaired, two-tailed) are applied for all statistical analyses in this figure. Values are expressed as the mean ± SEM. Significance was indicated as \*$p < 0.05$, \*\*$p < 0.01$, \*\*\*$p < 0.001$.

that phenylalanine inactivates insulin signaling through FARS, and that phenylalanylation underlies increased phenylalanine levels to inactivate insulin signaling.

We confirmed the role of phenylalanylation in the inactivation of insulin signaling in mice. As human FARS α subunit (hFARSA) overexpression resulted in more than 80% inactivation of insulin signaling (Supplementary Fig. 3c), FARSA was overexpressed in C57 mice (*hFARSA* mice) to examine the insulin signaling-inhibiting effects of FARS (Supplementary Fig. 3d). Although the food intake (Supplementary Fig. 3e) of *hFARSA* mice was comparable to that of the mother mouse strain, moderate body weight gain (Supplementary Fig. 3f), higher serum (Supplementary Fig. 3g) and hepatic (Supplementary Fig. 3h) triglyceride levels, and energy expenditure under comparable respiratory quotient (RQ) (Supplementary Fig. 3i) were observed in *hFARSA* mice fed standard chow. Blood glucose levels of *hFARSA* mice were significantly higher than those of the mother mouse strain starting four weeks after birth (Fig. 2g), and eight-week-old adult male *hFARSA* mice exhibited increased fasting blood glucose levels (Fig. 2h), reduced glucose tolerance (Supplementary Fig. 3j, 3k), reduced insulin tolerance (Fig. 2i), increased blood insulin levels (Fig. 2j), and increased HOMA-IR values (Fig. 2k), all of which phenocopied the consequences of phenylalanine chow feeding. Moreover, the adipose, liver, and muscle tissues of *hFARSA* mice showed decreased insulin signaling (Fig. 2l). Furthermore, insulin had a lower potency to stimulate the membrane localization of GLUT4 in *hFARSA* mouse muscle cells, in which GLUT4 was distributed mainly in the cytosol (Fig. 2m). These results are consistent with FARS-catalyzed phenylalanylation inhibiting insulin signaling.

## FARS phenylalanylated Lys1057 and Lys1079 of IR

To identify the proteins involved in insulin signaling that are regulated by phenylalanylation, we screened for how changes in phenylalanine and FARS levels affect the phosphorylation levels of proteins downstream of IR and IRS. Me-Phe supplementation phenocopied FARS overexpression and decreased insulin-induced phosphorylation levels of insulin signaling proteins, namely IRβ, AKT, and AS160 (Fig. 3a), as well as decreased phosphorylation levels of other insulin-activated IR targets, including SHC, ERK1/2, S6K, and GSK3β (Supplementary Fig. 4a, b), in 3T3-L1 cells. Moreover, IR knockout (*IR⁻/⁻*) in HepG2 cells abrogated Me-Phe supplementation-mediated decrease in insulin-induced phosphorylation of IRβ, AKT, and AS160, as observed in wild-type HepG2 cells (Fig. 3a). These results support the hypothesis that phenylalanylation acts on IR to regulate insulin signaling. Consistent with this notion, *IR* knockdown in 3T3-L1 cells, while blunting insulin to activate 2-DG uptake, abrogated both FARS overexpression and Me-Phe supplementation-mediated decrease in 2-DG uptake (Figs. 3b, c).

We identified two phenylalanylation sites in a tryptic peptide library of IRβ purified from 3T3-L1 cells. Lysine 1057 (K1057), an ATP-binding residue of IRβ[29], and lysine 1079 (K1079) were localized in the cytosolic-exposing domain of IRβ (Supplementary Fig. 5a) and were phenylalanylated (Supplementary Fig. 5b, 5c). The phenylalanylation of K1057 and K1079 was confirmed when synthetic IRβ peptides with either phenylalanylated K1057 or K1079 generated MS/MS spectra identical to those from the IRβ peptide library (Supplementary Fig. 5b,

c), and the levels of K1057 and K1079 phenylalanylation (F-K1057 and F-K1079) were increased by phenylalanine treatment in a dose-dependent manner (Fig. 3d) and by FARS overexpression (Fig. 3e), as detected with site-specific antibodies that recognized F-K1057- and F-K1079-containing IRβ peptides (Supplementary Fig. 5d, e).

FARSA co-localized with IR in HepG2 cells (Supplementary Fig. 6a). Moreover, FARSA co-immunoprecipitated with IR when co-expressed in HEK293T cells (Supplementary Fig. 6b). These results support the hypothesis that FARSA may phenylalanylate IR, given that physical interactions between ARSs and their substrates are prerequisites for ARSs to modify their substrates[21]. Purified FARSA, but not a phenylalanine binding-defective FARSA^Y412R/F438R mutant[30], phenylalanylated synthetic IRβ peptides containing K1057 and K1079 in the presence of FARSB (Fig. 3f) and increased F-K1057 and F-K1079 levels in purified intact IRβ in vitro (Fig. 3g). Moreover, the small interfering RNA (*siRNA*)-mediated reduction in FARSA expression in 3T3-L1 cells decreased F-K1057 and F-K1079 levels (Fig. 3h). Furthermore, F-K1057 and F-K1079 levels were high in the livers and muscles of *hFARSA* mice (Fig. 3i). These results confirm that FARS is a phenylalanyl transferase of K1057 and K1079 in IRβ. Furthermore, Me-Phe treatment at 2 mM in 3T3-L1 cells increased cellular phenylalanine (Supplementary Fig. 2a) and increased F-K1057 and F-K1079 levels from approximately 5% to 35% (Fig. 3j) as quantified with mass spectrometry. Consistently, aspartame treatment increased intracellular phenylalanine levels and F-K1057 and F-K1079 levels in a dose-dependent manner (Fig. 3k). These results suggest that the levels of F-K1057 and F-K1079 are substantially altered by phenylalanine level variation.

## Abrogating IR F-K1057/1079 prevented phenylalanine and FARS from inhibiting insulin signaling

To elucidate the importance of F-K1057 and F-K1079 in insulin signaling, we mutated either K1057 or K1079 or both residues to arginine (IRβ^K1057R, IRβ^K1079R, IRβ^2K/R) to mimic non-phenylalanylated IRβ, and we mutated either K1057 or K1079 or both residues to phenylalanine (IRβ^K1057F, IRβ^K1079F, IRβ^2K/F) to mimic phenylalanylated IRβ. When these mutants were each knocked in HepG2 cells and were expressed at comparable levels as those of wild-type IRβ, insulin had similar potency to increase IRβ 1150/1151 phosphorylation and activate insulin signaling in wild-type and non-phenylalanylation mimetic IRβ mutant knock-in HepG2 cells (Fig. 4a), but had a much weaker potency to activate insulin signaling in phenylalanylation mimetic IRβ mutant knock-in HepG2 cells (Fig. 4b). Moreover, IRβ^2K/R-expressing cells responded more strongly, and IRβ^2K/F-expressing HepG2 cells responded more weakly to insulin stimulation than wild-type IRβ-expressing cells (Supplementary Fig. 7a). IRβ^2K/R enabled stronger, and IRβ^2K/F enabled weaker, insulin responses than wild-type *IRβ*-silenced 3T3-L1 cells (Fig. 4c), consistent with insulin having much weaker potency to activate 2-DG uptake in IRβ^2K/F-expressing *IR* knockdown 3T3-L1 cells (Fig. 4d). These results suggest that phenylalanylation negatively regulates IR and insulin signaling.

Notably, Me-Phe supplementation failed to decrease the phosphorylation levels of indicators of insulin activation in both IRβ^2K/R- and IRβ^2K/F-knock in cells as it did in wild-type HepG2 cells (Supplementary Fig. 7b) and in simulated IRβ^2K/R- and IRβ^2K/F-expressing in *IRβ*-silenced

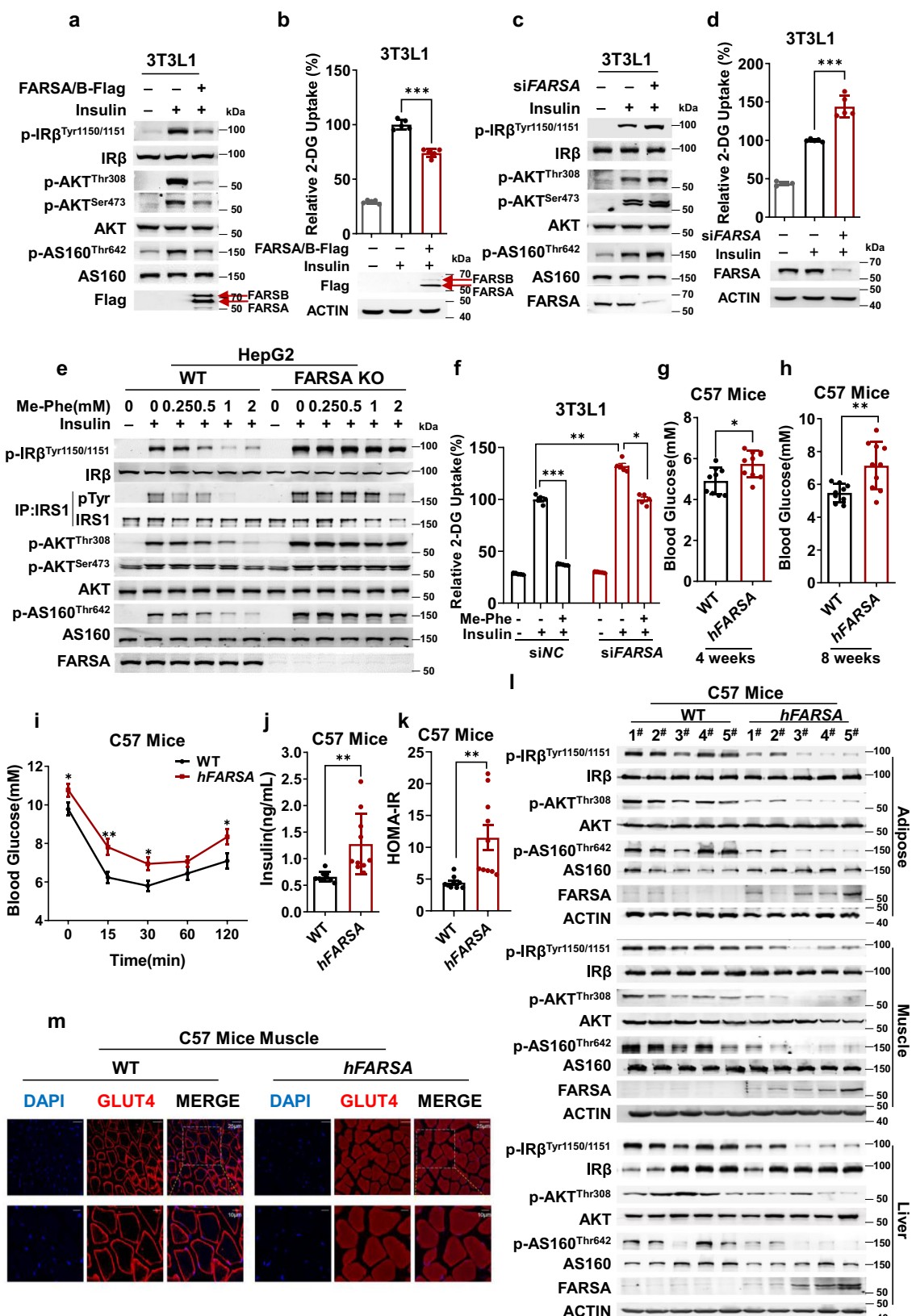

3T3L1 cells (Fig. 4e), consistent with the notion that Me-Phe supplementation is unable to decrease insulin-stimulated 2-DG uptake in both mutant IRβ-expressing IRβ knockdown 3T3-L1 cells (Fig. 4f). This suggests that K1057 and K1079 phenylalanylation mediates phenylalanine-induced insulin signaling inactivation. This notion is substantiated by the finding that FARS overexpression increased IRβ

phenylalanylation, inactivating insulin signaling in wild-type but not in IRβ²ᴷ/ᴿ- or IRβ²ᴷ/ᶠ-knock in HepG2 cells (Supplementary Fig. 7c), observations that were recaptured in *IRβ* knockdown 3T3-L1 cells, of which siRNA-resistant wild-type IRβ, but not IRβ²ᴷ/ᴿ or IRβ²ᴷ/ᶠ, inactivated insulin signaling (Fig. 4g) and decreased 2-DG uptake (Fig. 4h). Lastly, consistent with the fact that F-K1057 and F-K1079 underlie

**Fig. 2 | FARS overexpression phenocopies phenylalanine treatment efficacy.**
**a** FARSA/B overexpression impairs insulin signaling. Phosphorylation of insulin signaling components was detected in 3T3-L1 adipocytes and in 3T3-L1 adipocytes overexpressing both FARSA and FARSB (FARSA/B). **b** FARSA/B overexpression inhibits 2-DG uptake. The 2-DG uptake ability of 3T3-L1 adipocytes and 3T3-L1 adipocytes overexpressing FARSA/B was determined ($n = 5$). **c** *FARSA* silencing sensitizes insulin signaling. Phosphorylation of insulin signaling components was detected in 3T3-L1 adipocytes and 3T3-L1 adipocytes with *FARSA* knockdown using siRNA. **d** *FARSA* silencing increases 2-DG uptake by the cells. The 2-DG uptake ability of 3T3-L1 adipocytes and 3T3-L1 adipocytes with *FARSA* knockdown was analyzed ($n = 5$). **e** Me-Phe fails to affect insulin signaling in *FARSA*-knockout cells. The ability of Me-Phe to decrease the phosphorylation of insulin signaling components was tested in HepG2 cells and *FARSA*-knockout HepG2 cells. **f** *FARSA* silencing desensitizes phenylalanine to decrease 2-DG uptake. The ability of phenylalanine to inhibit 2-DG uptake was compared between 3T3-L1 adipocytes and *FARSA* knockdown 3T3-L1 adipocytes ($n = 5$). **g–k** *hFARSA* overexpression induces T2D symptoms in mice. Blood glucose at 4 weeks (**g**) and 8 weeks (**h**), insulin tolerance (**i**), and blood insulin levels (**j**) and HOMA-IR values (**k**) in fasted 8-week-old *hFARSA*-transgenic C57 mice were measured (WT, $n = 9$, hFARSA-transgenic, $n = 10$ mice). **l** *hFARSA* overexpression inhibits insulin signaling. Phosphorylation of components of the insulin signaling pathway was detected in the adipose, muscle, and liver tissues of C57 and *hFARSA*-transgenic C57 mice. **m** *hFARSA* overexpression blunts GLUT4 membrane translocation stimulated by insulin. The distribution of GLUT4 in the muscle tissues of C57 and *hFARSA*-transgenic C57 mice was detected after the same dose of insulin was injected. Significance was indicated as two-tailed, unpaired, $t$ test for (**b**, **d**, **g–k**), one-way ANOVA test for (**f**). Values are expressed as the mean ± SEM. *$p < 0.05$, **$p < 0.01$, ***$p < 0.001$.

insulin resistance, a high-fat diet, an established risk factor of insulin resistance[31], increased circulating phenylalanine levels (Supplementary Fig. 7d), decreased SIRT1 levels (Supplementary Fig. 7e) as reported[32,33], but had negligible effects on FARSA levels, consistently induced K1057/F-K1079 levels, and inhibited insulin signaling in the liver, muscle, and adipose tissues of mice (Supplementary Fig. 7e). However, treatment with the PI3K inhibitor wortmannin inhibited downstream insulin signaling in FARSA-knockout HepG2 cells, which had diminished F-K1057 and F-K1079 levels (Supplementary Fig. 7f), suggest that phenylalanylation on IRβ is sufficient but not necessary for inactivating insulin signaling.

### SIRT1 removed F-K1057/1079 and sensitized insulin signaling

We tested whether F-K1057 and F-K1079 were dynamically regulated. Given that sirtuins have deaminoacylation activitiy[21] and that the sirtuin inhibitor nicotinamide[34] rather than the histone deacetylase inhibitor trichostatin A (TSA) elevated F-K1057 and F-K1079 levels and inhibited insulin signaling (Fig. 5a), F-K1057 and F-K1079 were likely to be removed by sirtuins. Moreover, SIRT1, but not the other cytoplasmic SIRTs, namely, SIRT2, SIRT6, and SIRT7, decreased F-K1057 and F-K1079 levels and activated insulin signaling in HepG2 cells (Fig. 5b), suggesting that SIRT1 is a dephenylalanylase of F-K1057 and F-K1079.

Purified SIRT1 pulled down the cytoplasmic domain of IRβ in vitro (Supplementary Fig. 8a). Moreover, SIRT1 co-immunoprecipitated with IRβ when ectopically co-expressed in HEK293T cells (Supplementary Fig. 8b). Furthermore, recombinant SIRT1, but not the amidase-dead mutant SIRT1[H363Y][35], gave rise to NAD⁺-dependent dephenylalanylated synthetic F-K1057- and F-K1079-containing IRβ peptides (Fig. 5c) and intact IRβ purified from HepG2 cells (Fig. 5d) in vitro. These results confirm that SIRT1 is an F-K1057 and F-K1079 dephenylalanylase.

The notion that SIRT1 dephenylalanylates F-K1057 and F-K1079 was further substantiated by SIRT1's ability to regulate insulin signaling. SIRT1 overexpression decreased F-K1057 and F-K1079 levels and increased insulin signaling in hepatocytes of C57 mice (Fig. 5e). F-K1057 and F-K1079 levels were higher and insulin signaling was lower in hepatocytes of *Sirt1* KO C57 mice than in those of wild-type C57 mice, and the SIRT1-specific inhibitor EX527 was unable to alter insulin signaling in hepatocytes of *Sirt1* KO C57 mice but inhibited insulin signaling in hepatocytes of wild-type C57 mice (Fig. 5f). Me-Phe supplementation of the culture media of mouse hepatocytes increased F-K1057 and F-K1079 levels, inhibited insulin signaling, and decreased the potency of insulin to activate insulin signaling. Insulin desensitization effects were much more pronounced in hepatocytes of *Sirt1* KO mice than in those of wild-type mice (Fig. 5g). Moreover, ablation of the phenylalanylation sites of IRβ in HepG2 cells resulted in insulin signaling (Figs. 5h) and 2-DG uptake (Fig. 5i) in cells unresponsive to SIRT1 overexpression. These results are consistent with the notion that SIRT1 regulates insulin signaling by modulating F-K1057 and F-K1079 levels.

Elevated F-K1057 and F-K1079 levels, decreased insulin signaling (Fig. 5j), and reduced membrane distribution of GLUT4 (Fig. 5k) were also observed in *Sirt1* KO mouse muscles, the major organ involved in glucose disposal. These results are consistent with previous reports that *SIRT1* KO induces insulin resistance[36,37], which was confirmed by our findings that elevated fasting blood glucose (Supplementary Fig. 8c), reduced glucose tolerance (Supplementary Fig. 8d), insulin tolerance (Supplementary Fig. 8e), higher blood insulin levels (Supplementary Fig. 8f), and increased HOMA-IR values (Supplementary Fig. 8g) were observed in *Sirt1* KO mice.

### F-K1057 and F-K1079 levels were high in type 2 diabetes patients

To investigate whether IRβ F-K1057 and F-K1079 levels are associated with T2D onset, we collected and analyzed blood samples from 60 healthy individuals and 62 patients with T2D matched in age and gender (Supplementary Table 1). We first confirmed that the blood phenylalanine levels of T2D samples were higher than those of non-T2D subjects (Fig. 6a) and positively correlated with HbA1c, a key diabetes marker (Fig. 6b), confirming previous findings[11–13]. F-K1057 and F-K1079 levels and insulin signaling of human white blood cells (WBCs) were dose-dependently increased and inhibited, respectively, by phenylalanine (Fig. 6c), and the phenylalanine effects were recaptured in the WBCs of phenylalanine-fed mice (Fig. 6d), suggesting that WBCs may be used to detect phenylalanylation in clinical samples.

The levels of F-K1057 and F-K1079 in WBCs of patients with T2D were not only higher than those of the non-T2D subjects but also positively correlated with HbA1c levels (Fig. 6e–h), consistent with F-K1057 and F-K1079 levels underlying insulin signaling inhibition and insulin resistance. Moreover, SIRT1 protein levels, estimated by western blot analysis, were moderately lower in WBCs of T2D patients than in normal subjects (Fig. 6i), and negatively correlated with HbA1c levels (Fig. 6j), consistent with SIRT1 inactivation causing insulin resistance. However, FARSA protein levels were unexpectedly indistinguishable between patients with T2D and normal subjects (Fig. 6k), and there was no correlation between FARSA protein levels and HbA1c levels (Fig. 6l). These results suggest that WBC F-K1057 and F-K1079 levels are good clinical markers for T2D.

### Decreasing F-K1057 and F-K1079 levels restored insulin signaling

The effects of decreasing F-K1057 and F-K1079 levels on insulin signaling activation and glucose uptake were determined in cells and mice to confirm that F-K1057 and F-K1079 may cause T2D. Phenylalaninol is a structural analog of phenylalanine (Fig. 7a) that may inhibit F-K1057 and F-K1079 synthesis by occupying the phenylalanine binding pocket of FARSA (Fig. 7b, Supplementary Fig. 9a). This hypothesis was confirmed by the decrease in F-K1057 and F-K1079 levels in a dose-dependent manner and the activation of insulin signaling in HepG2 cells, but not in IRβ[2K/F] knock-in HepG2 cells (Fig. 7c), suggesting that phenylalaninol activates insulin signaling by inhibiting F-K1057 and

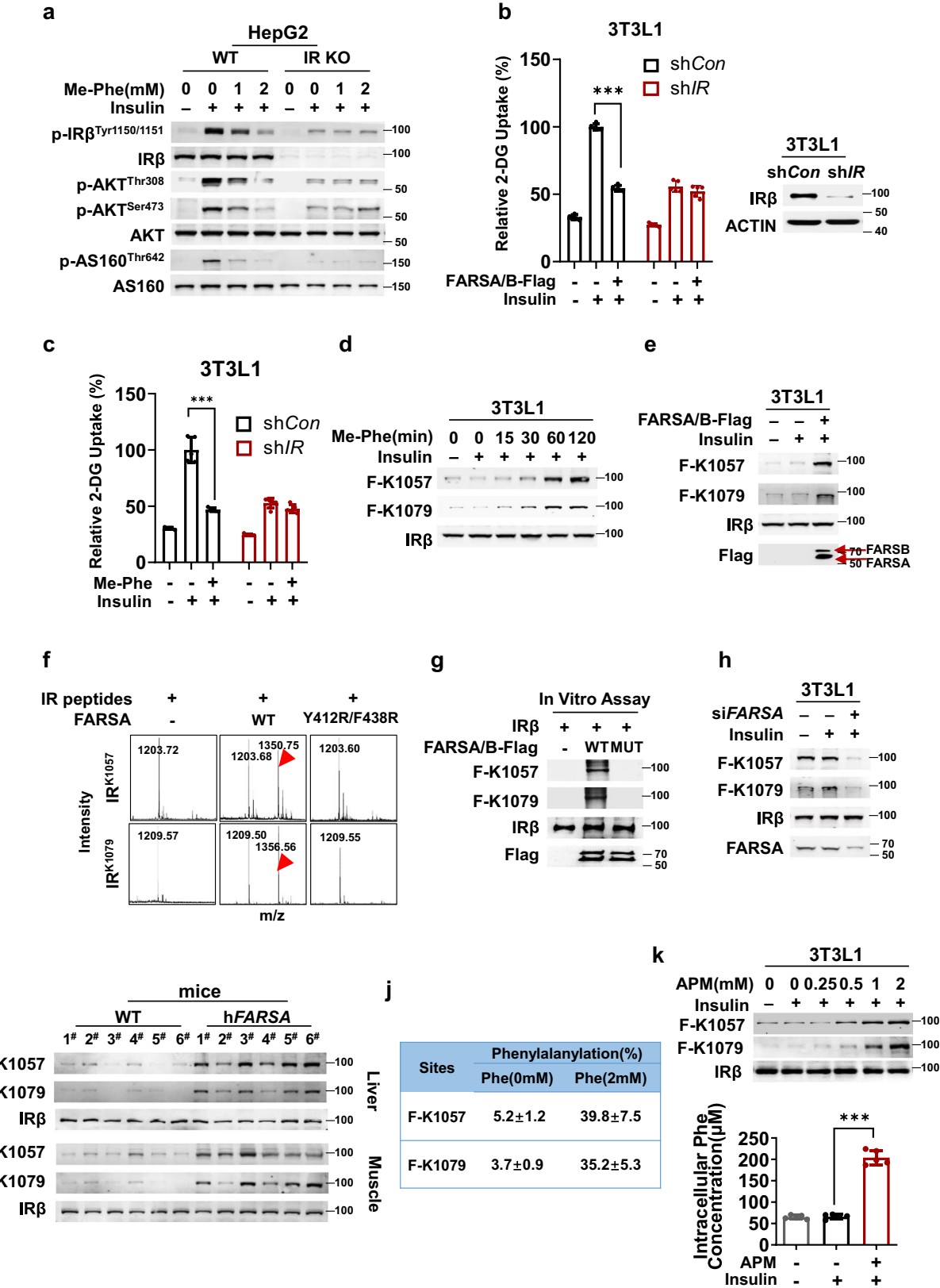

F-K1079 synthesis, which is consistent with phenylalaninol promoting 2-DG uptake by 3T3-L1 and L6 cells independent of insulin (Fig. 7d, Supplementary Fig. 9b).

Feeding mice with phenylalaninol-supplemented chow (PN chow) decreased the levels of F-K1057 and F-K1079 and increased phosphorylation levels of insulin signaling components in *hFASA* mouse livers (Fig. 7e, Supplementary Fig. 9c), and in the livers, adipose tissues, and muscles of aspartame chow-fed mice (Fig. 7f, Supplementary Fig. 10a). PN chow had no observable effects on food intake (Supplementary Fig. 3e), and moderately decreased serum and hepatic TG levels (Supplementary Fig. 3g, h), and energy expenditure (Supplementary Fig. 3i), consistent with phenylalaninol being an F-K1057 and

**Fig. 3 | FARS phenylalanylates IR Lys1057/1079. a** Me-Phe has negligible effects on insulin signaling in *IR*-KO HepG2 cells. Phosphorylation of the components of the insulin signaling pathway was detected in HepG2 and IR-KO HepG2 cells in the absence and presence of Me-Phe. **b** FARSA/B overexpression has no effect on 2-DG uptake in *IR*-silenced 3T3-L1 adipocytes. 2-DG uptake was detected in 3T3-L1 adipocytes and *IR*-silenced 3T3-L1 adipocytes (*n* = 5). **c** Me-Phe fails to inhibit 2-DG uptake in *IR*-silenced 3T3-L1 adipocytes. 2-DG uptake was detected in WT and *IR*-silenced 3T3-L1 adipocytes in the absence and presence of Me-Phe (*n* = 5). **d** Me-Phe increases F-K1057/1079 levels. The levels of F-K1057/1079 were detected in 3T3-L1 adipocytes at the indicated time points after the cells were treated with Me-Phe. **e** FARSA/B overexpression increases F-K1057/1079 levels. F-K1057/1079 levels were detected in 3T3-L1 adipocytes and in 3T3-L1 adipocytes overexpressing FARSA/B. **f** FARSA/B phenylalanylates IR K1057 and K1079 peptides. The ability of recombinant FARSA/B and FARSA^Y412R/F438R/B to phenylalanylate K1057-and K1079-containing IR peptides in vitro was tested. The formation of phenylalanylated

products (red arrows) was assayed by mass spectrometry. **g** FARSA/B phenylalanylates IRβ. The abilities of recombinant FARSA/B and FARSA^Y412R/F438R/B to phenylalanylate recombinant IR were tested. The formation of F-K1057/1079 was detected using western blotting. **h** *FARSA* knockdown reduces F-K1057/1079 levels. F-K1057/1079 levels were detected in 3T3-L1 adipocytes and *FARSA*-knockdown 3T3-L1 adipocytes. **i** hFARSA overexpression increases F-K1057/1079 levels. The levels of F-K1057/1079 were detected in the liver and muscle tissues of C57 and *hFARSA*-transgenic C57 mice. **j** Phenylalanine treatment increases F-K1057/1079 levels. The levels of F-K1057/1079 were quantified in 3T3-L1 adipocytes that were untreated or treated with phenylalanine. **k** Aspartame increases F-K1057/1079 levels. The levels of F-K1057/1079 were detected in 3T3-L1 adipocytes at the indicated doses after the cells were treated with aspartame (APM). Significance was indicated as two-tailed, unpaired, *t* test for (**k**), one-way ANOVA test for (**b**, **c**). Values are expressed as the mean ± SEM. *p < 0.05, **p < 0.01, ***p < 0.001.

---

F-K1079 synthesis inhibitor. Although PN chow did not affect caloric intake in mice (Supplementary Fig. 10b), it decreased body weight by 10% (Fig. 7g) and body fat (Fig. 7h) after 12 weeks of feeding. Moreover, the body weight- and body fat-reducing effects of phenylalaninol were more pronounced in *hFARSA* mice (Figs. 7g, 7h), which had higher body weights and body fat (Figs. 7g, 7h) than C57 mice. Furthermore, PN chow effectively reduced blood glucose levels in C57 and *hFARSA* mice (Fig. 7i). These results are consistent with the finding that F-K1057 and F-K1079 contribute to obesity, high blood glucose, and T2D phenotypes. The effects of phenylalaninol were also tested in C57BLKsJ-db/db mice (db/db mice). Although F-K1057 and F-K1079 levels in db/db mice were only slightly higher than those in C57 mice, PN chow feeding significantly reduced liver and muscle F-K1057 and F-K1079 levels in both strains (Fig. 7j) and resulted in enhanced glucose tolerance (Fig. 7k) and insulin tolerance (Fig. 7l), lowered blood insulin and glucose levels (Figs. 7m, n), decreased HOMA-IR values (Fig. 7o), and decreased body weight by 30% (Fig. 7p) after 6 weeks of feeding. These results suggest that F-K1057 and F-K1079 are major contributors to insulin resistance and that reversing them is beneficial for relieving T2D symptoms.

## Discussion

Using phenylalanine and aspartame to mimic extremes for serum Phe elevation in human samples, we demonstrate that FARS senses phenylalanine levels and converts them into a phenylalanine signal by modifying proteins. For IRβ, the phenylalanine signal F-K1057/1079 inactivates insulin signaling and inhibits glucose uptake, providing a protective mechanism of excess glucose uptake when intracellular amino acids are abundant. Therefore, phenylalanylation, together with already reported mechanisms including phosphorylation, sumolytation, myristylation, OGlc-NAc-ylation, ubiquitination of IRβ, provided another integrated response to nutrient excess. Moreover, SIRT1, a metabolic regulator that employs the cellular energy level indicator NAD+ as a cofactor, integrates cellular energy levels into insulin signaling and glucose uptake regulation. This elegant design prevents insulin signaling and glucose uptake activation when either high amino acid or cellular energy levels are detected by cells. A question that needs to be answered is whether every amino acid can signal its abundance. Interestingly, the most significantly elevated amino acids in patients with T2D are essential amino acids, including aromatic amino acids and branched-chain amino acids[11–13]. This suggests that essential amino acids are good indicators of amino acid abundance. It makes physiological sense because essential amino acids rely on uptake from the environment to fulfill cellular needs, and intracellular levels of essential amino acids are more accurate indicators of amino acid abundance than nonessential amino acids, which can be generated intracellularly from other sources. This is consistent with the

observation that fatty acids, palmitate, and myristate have more signaling roles than other fatty acids[38].

Although elevated phenylalanine and F-K1057/1079 or reduced SIRT1 levels are negatively correlated with insulin signaling activation and glucose uptake and positively correlated with T2D onset, consistent with SIRT1 ameliorating insulin resistance[39], no correlations were found between T2D onset and FARSA levels, a positive regulator of F-K^1057/1079, whose overexpression promoted insulin resistance in mice. This implies two possibilities. First, intracellular phenylalanine levels are the major determining factors of F-K^1057/1079 levels, and second, FARS overexpression is toxic and is under negative selection during evolution. Notably, inhibiting F-K1057/1079 levels significantly alleviated T2D symptoms in db/db mice, in which insulin resistance is not a direct consequence of phenylalanine overloading, highlighting the importance of F-K1057/1079 in T2D development and suggesting an alternative strategy to contain T2D. Of note, phenylalaninol seemed to moderately decrease mice body weights, which may complicate the explanation of T2D symptoms relieving by phenylalaninol in db/db mice, and ask further study to valid this manipulation. The limitations of the current study include that 1% in dietary exposure of Phe or aspartame to induce type 2 diabetes phenotypes is beyond real life exposures, and this may reduce the reliability of the conclusion. Second, we noticed that F-K1057/1079 slightly affect energy balance, as evidenced by altered weight gain and energy expenditure was found in overexpressing hFARSA tg and PN-fed mice (Supplementary Fig. 3e-i), which leaves a possibility that altered energy balance may also contribute to F-K1057/1079-mediated insulin sensitivity.

High-protein diets are often suggested for patients with T2D[40–42], despite studies indicating that prolonged high-protein intake induces insulin resistance[14,15,43]. High-protein diets, such as high-fat ketogenic diets, may have transient blood glucose-controlling effects but may impair insulin signaling after prolonged exposure and thus have mixed outcomes[44]. Moreover, aspartame is a widely used artificial sweetener that decreases sugar intake and is thought to be beneficial for T2D prevention. Our study suggests that, although aspartame is safe under the suggested doses, over-dosing or prolonged exposure may have adverse effects on insulin signaling, as previously reported[45,46].

## Methods
### Antibodies
Antibodies against IRβ rabbit (#3025, 1:1000), IRβ mouse (#3020, 1:500), p-IRβ^Tyr1150/1151 (#3024, 1:1000), IRS1(#2382, 1:1000), Pan-phospho-Tyrosine(#9461, 1:2000), AKT(#4685, 1:3000), p-AKT^Thr308 (#13038, 1:1000), p-AKT^Ser473 (#4060, 1:3000), AS160(#2670, 1:1000), p-AS160^Thr462 (#8881, 1:1000), SHC(#2432, 1:1000), p-SHC^Tyr239/240 (#2434, 1:1000), ERK1/2 (#9102, 1:1000), p-ERK1/2^Tyr202/204 (#9101, 1:1000), S6K(#2708, 1:1000), p-S6K^Thr389 (#9234, 1:1000), GSK3β(#12456, 1:1000), p-GSK3β^Ser9 (#5558, 1:1000), SIRT1(#9475, 1:1000) were purchased from Cell Signaling Technology (Danvers, Massachusetts, USA).

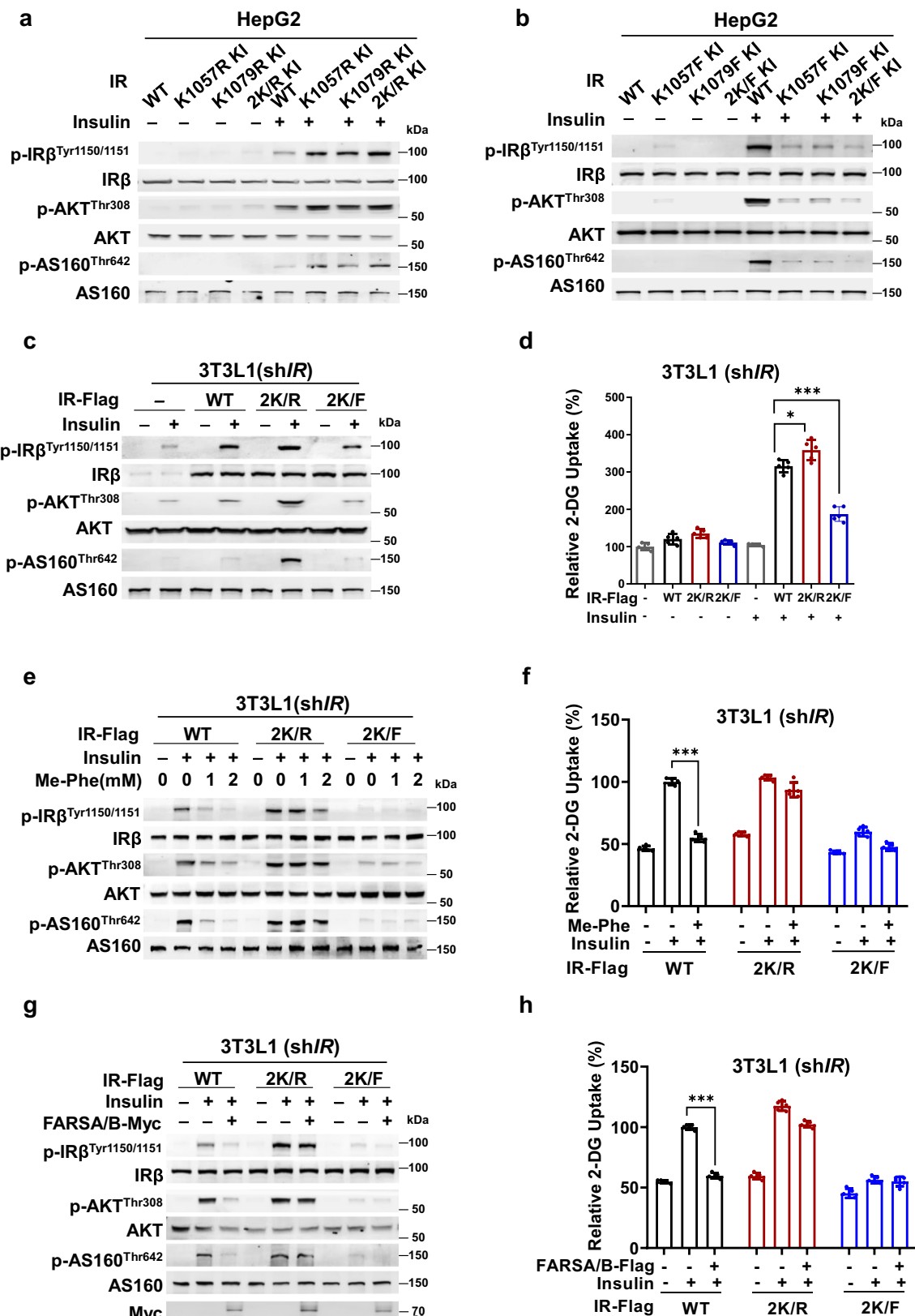

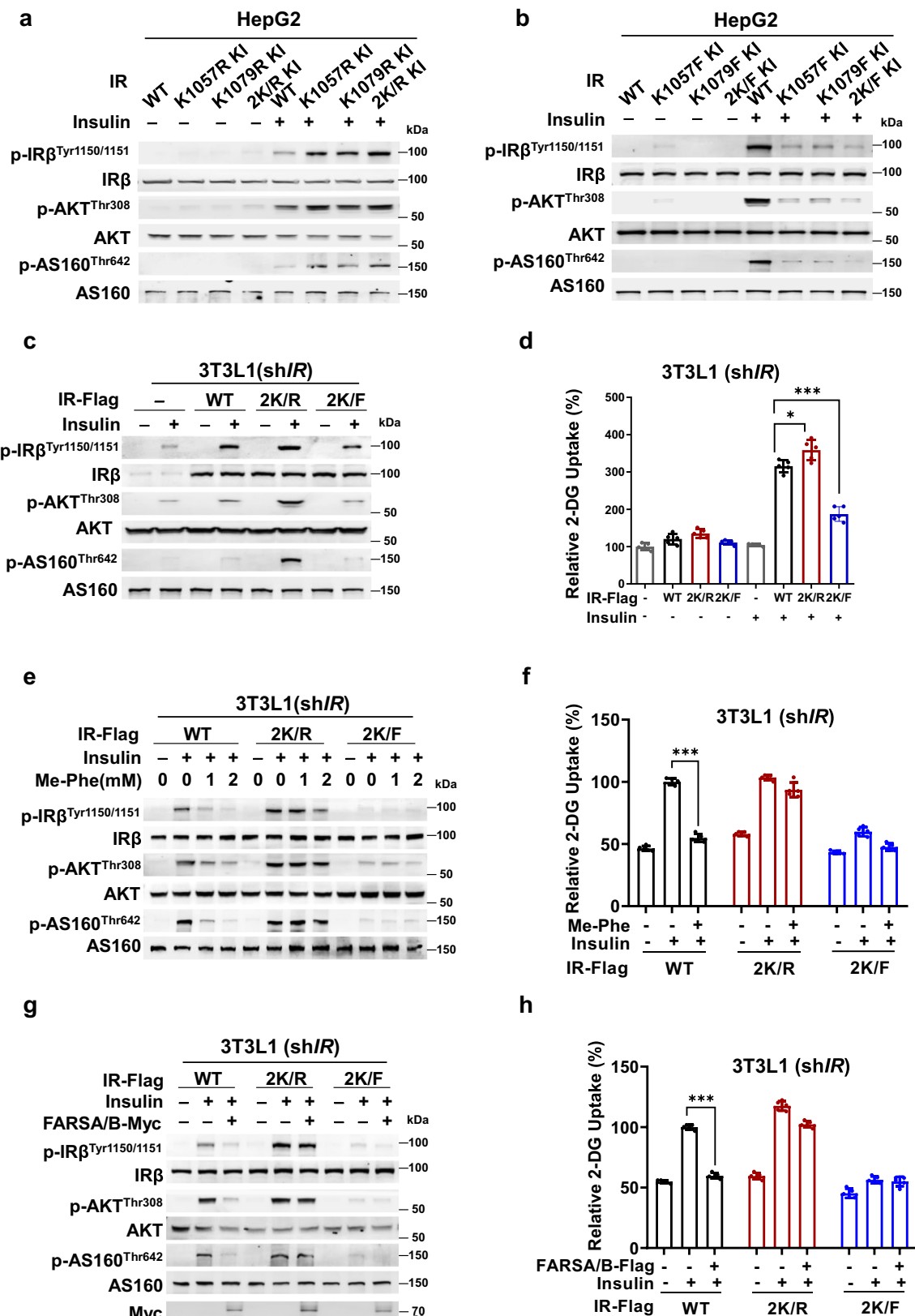

Antibodies against PAH (ab178430, 1:3000), IRβ (ab69508, 1:1000), GLUT4 (ab654, 1:3000) was purchased from Abcam (Cambridge, UK) . The antibodies against FARSA (18121-1-AP, 1:1000), SIRT1 mouse (60303-1-lg, 1:1000), FARS2 (16436-1-AP, 1:1000) was purchased from Proteintech (Rosemont, USA) . The antibody against Actin(#A00702, 1:5000), anti-rabbit secondary antibody and anti-mouse secondary

antibody were purchased from Genscript (Piscataway, USA) . The antibodies against Flag(#M20008, 1:3000), Myc(#M20002, 1:3000), HA(M20003, 1:3000) were purchased from Abmart (Berkeley Heights, USA) . Alexa Fluor Plus 488 donkey anti-mouse IgG secondary antibody (#A32766TR, 1:5000), Alexa Fluor Plus 555 donkey anti-rabbit IgG secondary antibody (#A32794TR, 1:5000), Alexa Fluor Plus 488 goat anti-

**Fig. 4 | Abrogating IR K1057/1079 phenylalanylation prevents phenylalanine and FARS from inhibiting insulin signaling. a, b** K1057/1079 phenylalanylation determines cell responses to insulin stimulation. The phosphorylation levels of components of the insulin signaling pathway were detected in HepG2 cells and HepG2 cells with either IRβ K1057, K1079, or both, and switched to non-phenylalanylation memetic arginine (**a**) or phenylalanylation memetic phenylalanine (**b**) with CRISPR-Cas9 in the absence and presence of insulin. **c, d** K1057/1079 phenylalanylation inactivates IRβ and inhibits glucose uptake. The effects of insulin on the phosphorylation levels of the components of the insulin signaling pathway (**c**) and 2-DG uptake ($n = 5$) (**d**) were detected *in IR* knockdown 3T3-L1 adipocytes with IRβ, IRβ$^{2K/R}$, and IRβ$^{2K/F}$ expressed at comparable levels. **e, f** Absence of K1057/ K1079 abrogates Me-Phe to alter insulin signaling and glucose uptake. The responses of insulin signaling (**e**) and 2-DG uptake ($n = 5$) (**f**) to Me-Phe treatment were detected in *IR* knockdown 3T3-L1 adipocytes treated with IRβ, IRβ$^{2K/R}$, and IRβ$^{2K/F}$ expressed at comparable levels. **g, h** FARS overexpression fails to inhibit insulin signaling and glucose uptake when K1057/K1079 is absent in 3T3-L1 cells. The responses of insulin signaling (**g**) and 2-DG uptake ($n = 5$) (**h**) to FARSA/B overexpression were detected in *IR* knockdown 3T3-L1 adipocytes expressing IRβ, IRβ$^{2K/R}$, and IRβ$^{2K/F}$, respectively. One-way ANOVA are applied for all statistical analyses in this figure. Values are expressed as the mean ± SEM. Significance was indicated as \*$p < 0.05$, \*\*$p < 0.01$, \*\*\*$p < 0.001$.

rabbit IgG secondary antibody (#A11008, 1:5000) were purchased from Invitrogen (California, USA).

The Anti-F-K1057 (1:500) and Anti-F-K1079 (1:500) antibodies were generated by Abmart Shanghai Co., Ltd during this study. Briefly, synthetic peptides (KGEAETRVAVK$_{1057Phe}$TVNESASLRE) and IEFL-NEASVMK$_{1079Phe}$GFTCHHVVR) were conjugated to keyhole limpet hemocyanin (KLH) as antigen. Rabbits were immunized by four doses of subcutaneous injection at two weeks interval between each injection before the rabbits were sacrificed for anti-sera. Antibodies were immunoaffinity purified by antigen and tested for specificities by blot assay before employed for western blotting analysis.

## Plasmids constructs and transfection

Whole length human FARSA, FARSB, FARS2, IR, SIRT1, SIRT2, SIRT6, and SIRT7 were amplified from HEK293T cDNA and cloned into the Xho I and EcoR I restriction sites of the pcDNA3.1-Flag/HA vector using CloneExpress MultiS One Step Cloning Kit (#C113-02, Vazyme, Nanjing, China). The IR mutants were generated by site-directed mutagenesis using the Mut Express MultiS Fast Mutagenesis kit (#C215-01, Vazyme) according to the manufacturer's instructions. Each plasmid was transfected using Lipofectamine 3000 (Invitrogen, Carlsbad, USA) according to the manufacturer's instructions.

The primers used were as follows:

FARSA: forward, 5′-AAC GGG CCC TCT AGA CTC GAG ATG GCG GAT GGT CAG-3′ reverse, 5′-TAG TCC AGT GTG GTG GAA TTC CGC AGC CTC CTG TGT-3′

FARSB: forward, 5′-AAC GGG CCC TCT AGA CTC GAG ATG CCG ACT GTC AGC GTG-3′ reverse, 5′-TAG TCC AGT GTG GTG GAA TTC CAA AAA GGG TCC AAC ATT −3′

FARS2: forward, 5′-AAC GGG CCC TCT AGA CTC GAG ATG GTG GGC TCA GCT CTC-3′ reverse, 5′-TAG TCC AGT GTG GTG GAA TTC GAA CCT GCC CTC CAC ACC −3′

IR: forward, 5′-AAC GGG CCC TCT AGA CTC GAG ATG GCC ACC GGG GGC CGG-3′ reverse, 5′-TAG TCC AGT GTG GTG GAA TTC GGA AGG ATT GGA CCG AGG-3′

SIRT1: forward, 5′-AAC GGG CCC TCT AGA CTC GAG ATG GCG GAC GAG GCG GCC-3′ reverse, 5′-TAG TCC AGT GTG GTG GAA TTC TGA TTT GTT TGA TGG ATA G-3′

SIRT1 H363Y: forward, 5′-GGA ATC CAA AGG ATA ATT CAG TGT CAT GGT TCC TT-3′ reverse, 5′-GCA AGA TGC TGT TGC AAA GGA ACC ATG ACA CTG AA −3′

SIRT2: forward, 5′- AAC GGG CCC TCT AGA CTC GAG ATG GAC TTC CTG CGG AAC T-3′ reverse, 5′-TAG TCC AGT GTG GTG GAA TTC CTG GGG TTT CTC CCT CTC T-3′

SIRT6: forward, 5′- AAC GGG CCC TCT AGA CTC GAG ATG TCG GTG AAT TAC GCG G-3′ reverse, 5′- TAG TCC AGT GTG GTG GAA TTC GCT GGG GAC CGC CTT GGC C-3′

SIRT7: forward, 5′- AAC GGG CCC TCT AGA CTC GAG ATG GCA GCC GGG GGT CTG-3′ reverse, 5′- TAG TCC AGT GTG GTG GAA TTC CGT CAC TTT CTT CCT TTT GT-3′

IR K1057R: forward, 5′-G GCG GTG AGG ACG GTC AAC GAG TCA GCC AGT C-3′ reverse, 5′-T GAC CGT CCT CAC CGC CAC GCG GGT CTC TGC C-3′

IR K1057F: forward, 5′-G GCG GTG TTC ACG GTC AAC GAG TCA GCC AGT C-3′ reverse, 5′-T GAC CGT GAA CAC CGC CAC GCG GGT CTC TGC C-3′

IR K1079R: forward, 5′-G GTC ATG AGG GGC TTC ACC TGC ATC AC GTG G-3′ reverse, 5′-T GAA GCC CCT CAT GAC CGA GGC CTC ATT GAG G-3′

IR K1079F: forward, 5′-G GTC ATG TTC GGC TTC ACC TGC ATC AC GTG G-3′ reverse, 5′-T GAA GCC GAA CAT GAC CGA GGC CTC ATT GAG G-3′.

## Cell lines and treatments

**Cell lines.** HEK293T cells and HepG2 cells were cultured in Dulbecco's Modified Eagle's Medium (DMEM) (Gibico) supplemented with 10% fetal bovine serum (FBS) (Gibico, Carlsbad, USA), 100 units/ml penicillin (Invitrogen, Carlsbad, USA) and 100 mg/ml streptomycin (Invitrogen, Carlsbad, USA).

3T3-L1 fibroblasts were cultured and differentiated into adipocytes by established protocols[47]. In brief, 3T3-L1 fibroblasts were cultured in DMEM supplemented with 10% newborn calf serum (NCS). Two days after confluence, cells were switched into differentiation medium containing 10% FBS, 1 mM insulin, 0.5 mM 3-isobutyl-1-methylxanthine and 1 mM dexamethasone for 2 days. The differentiation medium was then replaced with media containing 10% FBS and 167 nM insulin for another 2 days. Cells were then maintained in DMEM with 10% FBS. Adipocytes were used for experiments at day 10 post differentiation.

L6 rat skeletal myoblasts were cultured in DMEM with 10% FBS and antibiotics. After cultures reached 80%–90% confluency, medium was switched to 2% FBS to allow differentiation and myotube formation. After 8–10 days, cultures were used for experiments.

Primary hepatocytes isolation was performed according to published methods[48]. Briefly, D-Hanks' buffer (Calcium and magnesium-free Hanks' Balanced Salt Solution (HBSS) (Gibico, Carlsbad, USA) that contained 0.2% BSA and Hanks' buffer (HBSS, calcium and magnesium containing 0.2% BSA and collagenase 0.5 mg/ml (Gibico, Carlsbad, USA) was prewarmed at 40 °C in a water bath. Mice were anaesthetized by intramuscular injection of pentobarbital sodium. Hepatocytes were prepared by collagenase digestion via catheterization of the inferior vena cava (IVC) using a 24 G needle catheter. Prior to collagenase infusion, the liver was perfused (3-4 min) with D-Hanks' buffer via IVC after severing the portal vein. The color of the liver was observed, which changed to a beige or light brown color after perfusion. The liver was then perfused with collagenase in Hanks' buffer until the appearance of cracking on the liver surface. Collagenase perfused livers were excised out into DMEM and cells from the digested livers were teased out, suspended in DMEM, filtered through 70 μm nylon filter and centrifuged at 60×g for 6 min. The cell pellet was then mixed with Percoll (adjusted to physiological ionic strength with 10×PBS) to a final concentration of 30% and centrifuged at 2500 rpm for 10 min. Hepatocytes were collected and then cultured on collagen-coated plates (100.000 cells per well in 24-well plate) in Williams Medium E (Life Technologies, Carlsbad, USA) containing penicillin and streptomycin, 10 nM dexamethasone (all from Life Technology, Carlsbad, USA) and 10% FBS, before they can be cultured in serum-free medium.

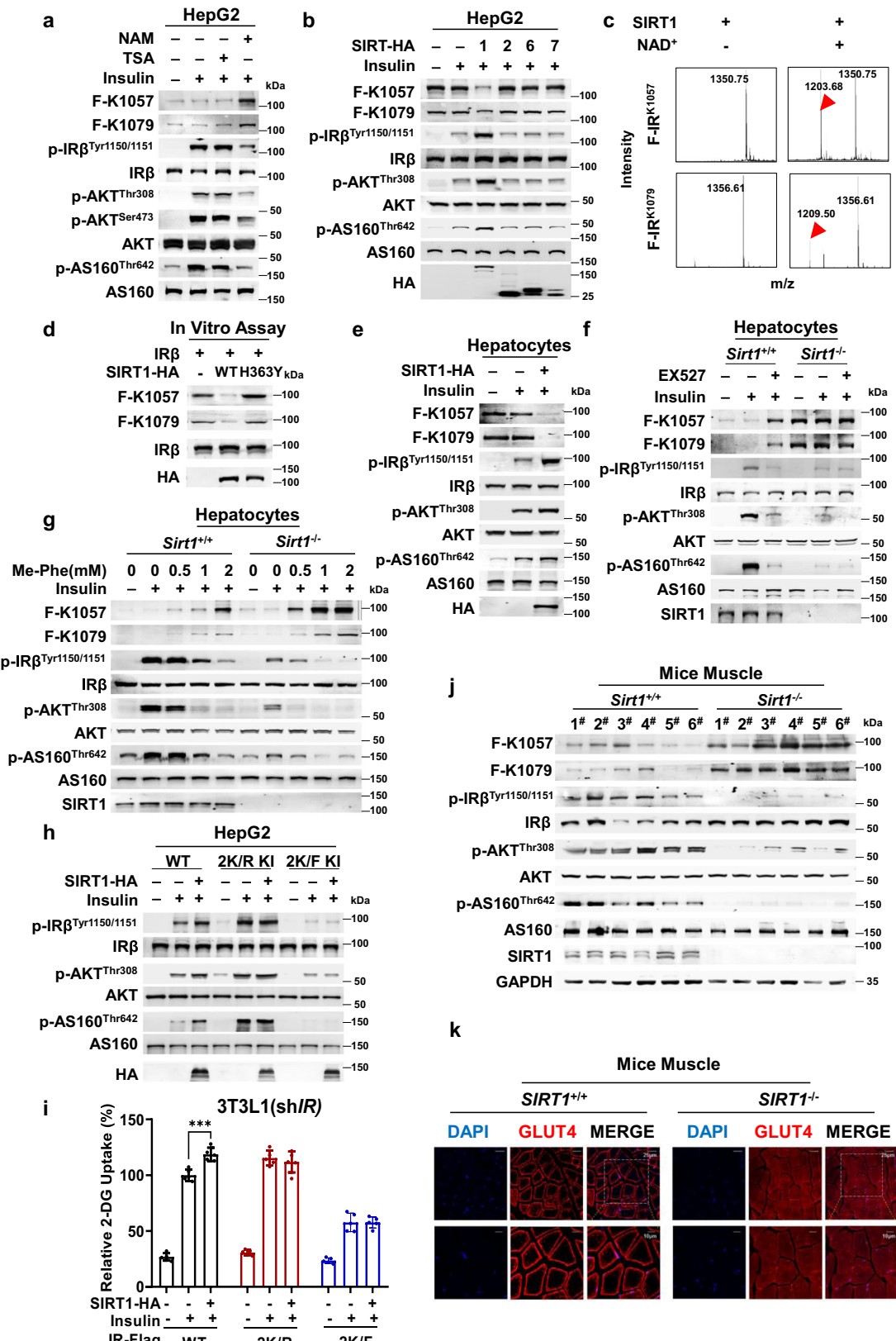

**Cell treatments.** Methyl-phenylalanine or aspartame treatment to cells was achieved by incubating cells in phenylalanine-free RPMI 1640 (Gibico, Carlsbad, USA) without serum for 60 min, followed by treatment with Methyl-phenylalanine or aspartame for 15–120 min.

Phenylalaninol treatment to cells was achieved by incubating cells in serum-free DMEM for 2 h, followed by treatment with indicated

phenylalaninol for 2 h. To measure the inhibiting effects of phenylalanine or FARSA to insulin signaling, 100 nM insulin was used to stimulate insulin signaling for 5 min before harvest of cells.

Trichostatin A treatment was carried out by adding 10 µM TSA to the culture medium 16-20 h before harvesting cells. Nicotinamide treatment was carried out by adding 5 mM nicotinamide to the culture

**Fig. 5 | SIRT1 dephenylalanylates K1057/1079 phenylalanylation and sensitizes insulin signaling. a** Sirtuin inhibition elevates F-K1057 and F-K1079 levels and inactivates insulin signaling. F-K1057/1079 levels and phosphorylation of components of insulin signaling were detected in HepG2 cells that were untreated and treated with NAM (5 mM) and TSA (0.5 μM). **b** SIRT1 decreases F-K1057/1079 levels and activates insulin signaling. F-K1057/1079 levels and phosphorylation of components of insulin signaling were detected in HepG2 cells and HepG2 cells overexpressing SIRT1, SIRT2, SIRT6, and SIRT 7, respectively. **c** SIRT1 removes F-K1057 and F-K1079 in vitro. Recombinant SIRT1 was incubated with synthetic F-K1057- and F-K1079-containing IRβ peptides in deacetylation buffer and the resulting peptides were analyzed by MS. **d** SIRT1 dephenylalanylates intact IRβ via deacetylase activity. Recombinant SIRT1 and deacetylase-inactivated SIRT1$^{H363Y}$ were incubated with IRβ in deacetylation buffer, and the levels of F-K1057/1079 were detected. **e** SIRT1 overexpression decreases F-K1057/1079 levels and activates insulin signaling. F-K1057/1079 levels and phosphorylation levels of components of the insulin signaling pathway were detected in mouse primary hepatocytes and mouse primary hepatocytes overexpressing SIRT1. **f** SIRT1 inhibitor fails to alter F-K1057/1079

levels and insulin signaling in Sirt1$^{-/-}$ mouse primary hepatocytes. Effects of EX527 (10 μM) on F-K1057/1079 levels and phosphorylation levels of components of the insulin signaling pathway were detected in wild-type and Sirt1$^{-/-}$ C57 mouse primary hepatocytes. **g** SIRT1$^{+/+}$ and SIRT1$^{-/-}$ mouse hepatocytes (F-K1057/1079) respond differently to phenylalanine. F-K1057/1079 levels and insulin-stimulated insulin signaling responses to phenylalanine were detected in SIRT1$^{+/+}$ and SIRT1$^{-/-}$ mouse primary hepatocytes. **h, i** Absence of K1057/1079 abrogates SIRT1-mediated activation of insulin signaling and 2-DG uptake. The ability of SIRT1 overexpression to activate insulin signaling (**h**) in wild-type, IRβ$^{2K/R}$, and IRβ$^{2K/F}$ knock-in HepG2 cells, and 2-DG uptake (n = 5) (**i**) in IR knockdown 3T3-L1 adipocytes expressing IRβ, IRβ$^{2K/R}$, and IRβ$^{2K/F}$ was assessed. Significance was indicated as one-way ANOVA. Values are expressed as the mean ± SEM. *$p < 0.05$, **$p < 0.01$, ***$p < 0.001$. **j** Sirt1 knockout increases F-K1057/1079 levels and impairs insulin signaling. F-K1057/1079 levels and phosphorylation of components of insulin signaling were detected in the muscles of wild-type and Sirt1$^{-/-}$ mice. **k** Sirt1 knockout decreases membrane enrichment of GLUT4. Cellular GLUT4 distribution was detected by immunofluorescence in the muscles of wild-type and Sirt1$^{-/-}$ mice.

medium 4-8 h followed by incubating cells in serum-free DMEM for 2 h before harvesting cells.

SIRT1-specific inhibitor EX527 treatment was carried out by adding 10 μM EX527 to the culture medium 24 h, followed by incubating cells in serum-free DMEM for 2 h before harvest cells.

PI3K inhibitor wortmannin treatment was achieved by incubating cells in serum-free DMEM for 2 h, followed by adding 0.2 or 1 μM wortmannin to the culture medium for 1 h, 100 nM insulin was used to stimulate insulin signaling for 5 min before harvest of cells.

## Cell lines genetic manipulations

**Small RNA interference.** Synthetic oligos were used for siRNA-mediated silencing of FARSA/FARS2/PAH, and scramble siRNA was used as a control. Cells were transfected with siRNAs using Lipofectamine 3000 (Invitrogen, Carlsbad, USA) according to the manufacturer's protocol. Knockdown efficiency was verified by western blotting. siRNA sequences was as follows: FARSA (mouse): 5′-TTG GCA GTG ACC TAC TAT TTA TT-3′, FARS2 (mouse): 5′-GCG ATC CAG GAC ACC TCT ATT TT-3, PAH (mouse): 5′-CGA AAG CAG TTT GCT GAC ATT TT.

**IR stable knockdown cell line.** HEK293T cells were co-transfected with pCMV-VSV-G, pCMV-Gag-Pol, and pMKO-IR. Transfected cells were cultured in DMEM containing 10% FBS for 6 h. 24 h after transfection, culture medium supernatant was collected and used for retrovirus preparation to infect 3T3L1 adipocytes at 10% confluency in 90-mm-diameter dishes. Cells were re-infected 24 h after the first infection and selected using 5 μg/mL puromycin (Amresco, Solon, OH, USA). Mouse IR shRNA was cloned into the Age I and EcoR I restriction sites of the pMKO vector. The sequences of used primers were: shIR forward, 5′-CCG GCC CTG AAG GAT GGA GTC TTT ACT CGA GTA AAG ACT CCA TCC TTC AGG GTT TTT G-3′, shIR reverse, 5′-AAT TCA AAA ACC CTG AAG GAT GGA GTC TTT ACT CGA GTA AAG ACT CCA TCC TTC AGG G-3′.

**CRISPR/Cas9 genomic knock-in cell lines.** IR K1057/1079F, IR K1057/1079R knock-in cell lines were all generated by using CRISPR/Cas9 mediated-mutagenesis as described previously[49]. Briefly, sgRNA-pX458 and ssODNs were when co-transfected into 70-80% confluent HepG2 cells by using Lipofectamine 3000 (Invitrogen). pX458 was transfected into 70-80% confluent HepG2 cells as negative control. 24 h after transfection, FACS selection was performed to sort GFP-positive single cells into the 96-well plates and then allowed them to expand for 2–3 weeks under standard culture condition in cell-incubator. PCR analysis was performed for cells after they grew to 60-70% confluency and the success of knockin was verified by both Sanger sequencing of genomic DNA and western blotting for cell lysate.

The guide sequences targeting the IR K1057 were:
Forward, 5′-CACCGTGACCGTCTTCACCGCCACG-3′
Reverse, 5′-AAACCGTGGCGGTGAAGACGGTCAC-3′
The guide sequences targeting the IR K1079 were:
Forward, 5′-CACCGGTGAAGCCCTTCATGACCG-3′
Reverse, 5-AAACCGGTCATGAAGGGCTTCACC-3′.

**CRISPR/Cas9 generation of knockout cells.** To generate FARSA and IR knockout HepG2 cells, the following guide sequence targeting the human FARSA forward: 5′-CAC CGC CGC CAA CTC GGC GCT GTC C-3′, reverse: 5′-AAA CGG ACA GCG CCG AGT TGG CGG C-3′, and the human IR forward: 5′-CAC CGG CGG TGG CCG CGC TGC TAC T-3′, reverse: 5′-AAA CAG TAG CAG CGC GGC CAC CGC C-3′ were used, following standard CRISPR/Cas9 gene editing protocols.

## Mice

All animal procedures were in accordance with the animal care committee at Fudan University.

**Mice maintenance and treatments.** Male C57BL/6J mice about six-week-old were obtained from Shanghai SLAC Laboratory Animal Co.,Ltd. (Shanghai, China). Male db/db mice were purchased from Gem Pharmatech Co., Ltd. (Nanjing, Jiangsu). Mice were housed in cages under a 14 h light/14 h dark cycle in a temperature- and humidity-controlled room, and freely fed standard laboratory chow with water ad libitum.

Mice were randomly divided into control and experimental groups when testing the phenylalanine or phenylalaninol efficacies. The control was fed with standard chow (Research Diet, AIN93G) and the experimental group was fed with either 1% or 0.1% phenylalanine chow or 1% phenylalaninol chow (modified from AIN-93G, Research Diet) that were manufactured from Shanghai SLAC Laboratory Animal Co., Ltd for 12 weeks.

Mice were randomly divided into control and experimental groups when testing the aspartame efficacies. The control was fed with standard chow (Research Diet, AIN93G) and the experimental group was fed with either 1% aspartame chow or 1% aspartame plus phenylalaninol chow (modified from AIN-93G, Research Diet)that were manufactured from Shanghai SLAC Laboratory Animal Co., Ltd for 12 weeks. Food intake measurements followed published protocol[50].

In high-fat diet animal models, male C57 mice freely fed with the standard laboratory chow (Research Diet, D12450) and the high-fat diet chow (Research Diet, D12492).

**Transgenic mice.** The hFARSA-transgenic mice were generated in a C57BL/6 genetic background by inserting CAG-LSL-hFARSA-WPRE-PA

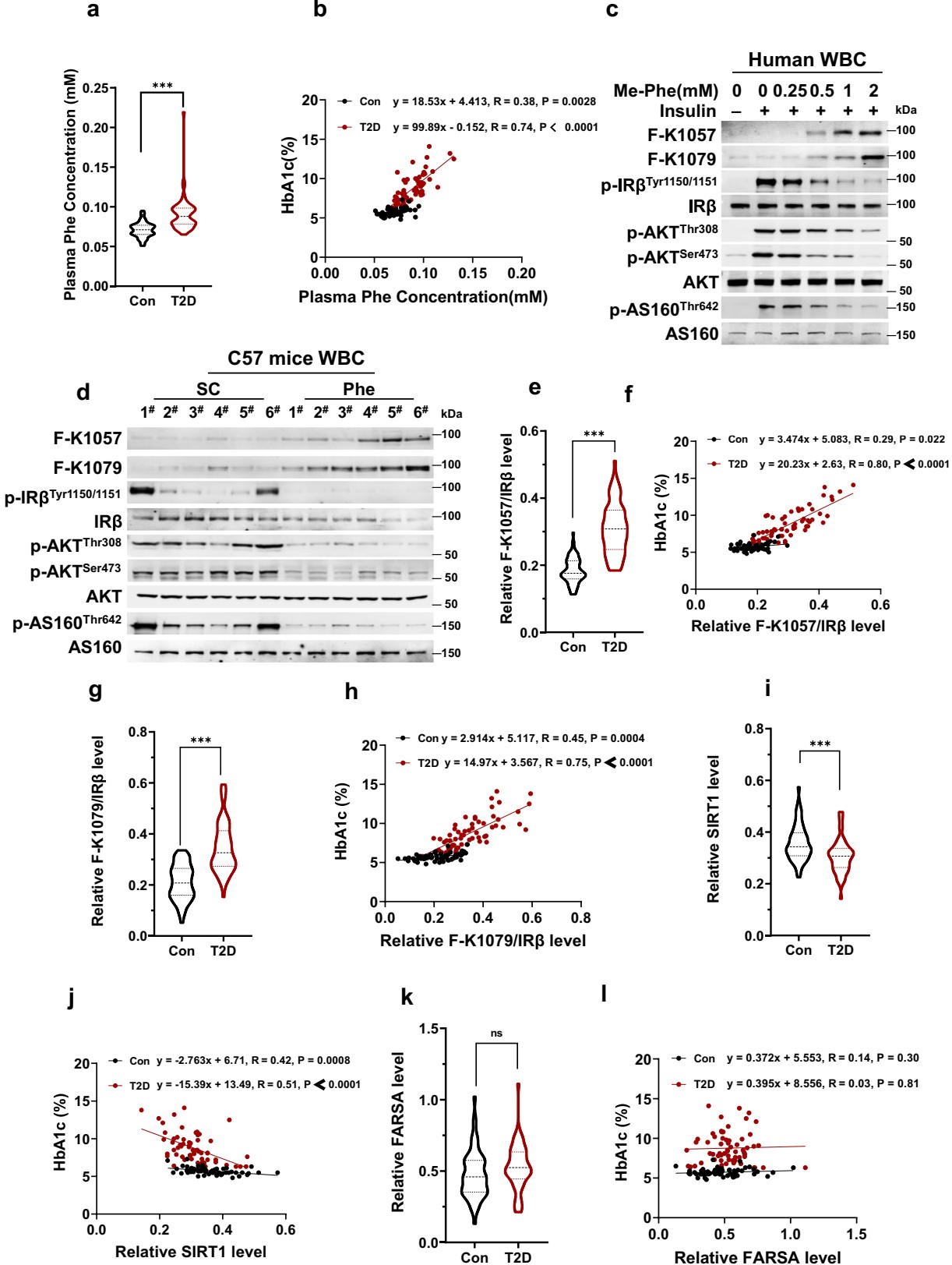

in Gt (ROSA) 26S or (ENSMUSG00000086429) gene using a CRISPR-Cas9-mediated genome editing system by Shanghai Model Organisms Center, Inc. Cas9 mRNA was in vitro transcribed with mMESSAGE mMACHINET7 Ultra Kit (Ambion, TX, USA) according to the manufacturer's instructions, and subsequently purified using the MEGA-clearTM Kit (Thermo Fisher, USA). 5'-GGGGACACACTAAGGGAGCT-3'

was chosen as Cas9 targeted guide RNA (sgRNA) and in vitro transcribed using the MEGAshortscript Kit (Thermo Fisher, USA) and subsequently purified using MEGAclearTM Kit. The donor vector with sgRNA and Cas9 mRNA was microinjected into C57BL/6J fertilized eggs. F0 generation mice positive for homologous recombination were identified by long PCR. The PCR products were further confirmed

**Fig. 6 | Elevation of F-K1057/1079 levels in patients with T2D. a** Phenylalanine levels are elevated in plasma samples from patients with T2D. Phenylalanine levels were measured in plasma samples from patients with T2D ($n = 62$, red spots, hereafter in this figure) and matched control samples ($n = 60$, black spots, hereafter in this figure). **b** Phenylalanine levels are positively associated with HbA1c levels in plasma samples from patients with T2D. Phenylalanine levels in patients with T2D and healthy subjects were plotted against their corresponding HbA1c levels. **c** F-K1057/1079 levels and insulin signaling are inversely regulated in human WBCs. F-K1057/1079 levels and insulin signaling were detected in human WBCs treated with the indicated levels of phenylalanine in the culture media. **d** Phe-fed mice have higher F-K1057/1079 levels and lower insulin signaling. F-K1057/1079 levels and insulin signals were detected in Phe-fed and normal chow-fed mice ($n = 6$). **e**–**h** F-K1057/1079 levels are correlated with T2D. F-K1057 (**e**) and F-K1079 levels (**f**) were compared between patients with T2D and matched control samples, and F-K1057 (**g**) and F-K1079 levels (**h**) were plotted against their corresponding HbA1c levels. **i, j** SIRT1 protein levels are low in patients with T2D. SIRT1 levels in patients with T2D and healthy subjects were measured (**i**) and their values were plotted against their corresponding HbA1c levels (**j**). **k, l** FARSA protein levels are negligibly altered in patients with T2D. The FARSA levels of patients with T2D and healthy subjects were measured (**k**), and their values were plotted against their corresponding HbA1c levels (**l**). Student's $t$ tests (unpaired, two-tailed) are applied for all statistical analyses in this figure. Values are expressed as the mean ± SEM. Significance was indicated as *$p < 0.05$, **$p < 0.01$, ***$p < 0.001$.

by sequencing. F0 mice were crossed with C57BL/6J mice to obtain h*FARSA* heterozygous mice.

*Sirt1*^flox/flox^ mice (C57BL/6-*Sirt1*^tm1(flox)Smoc^, containing two loxP sites flanking exon 3) was obtained from Shanghai Model Organisms Center, Inc. (Shanghai, China). The Sirt1 knockout mice were created by cross-breeding *Sirt1*^flox/flox^ and UBC-cre/ERT2 mice. The resulting UBC-cre/ERT2 *Sirt1*^flox/flox^ mice (Jackson Stock Number #007001) were mated with *Sirt1*^flox/flox^ mice to obtain Sirt1 conditional knockout mice (Sirt1 cKO mice; *Sirt1*^−/−^) and the corresponding *Sirt1*^flox/flox^ mice (*Sirt1*^+/+^) as the knockout control. Cre activity was induced by intraperitoneal injection of tamoxifen (100 mg/kg) daily for a total of 7 days. Genotyping was performed by PCR analysis of tail DNA. The primer sequences used to detect UBC-cre/ERT2 transgene were as follows, UBC-transgene: forward 5′-GAC GTC ACC CGT TCT GTT G-3′, reverse 5′-AGG CAA ATT TTG GTG TAC GG-3′, UBC-WT: forward 5′-CTA GGC CAC AGA ATT GAA AGA TCT-3′, reverse 5′-GTA GGT GGA AAT TCT AGC ATC ATC C-3′. Primer sequences used for genotyping *Sirt1*^flox/flox^ were as follows: forward 5′-GGA GCT GGG GTA TGT AAG ACG CAG AAA AA-3′, reverse 5′-TGG CCT ACA TCT GAC TAA CTC AAA TCC CT-3′.

## Clinical samples

**Participants.** Human IR K1057/K1079 phenylalanylation studies were conducted using fresh clinical Blood samples of T2D patients and matched healthy subjects were obtained from volunteers of Huashan hospital, Shanghai with known age, height, fasting blood glucose, insulin, HbA1c, triglyceride, cholesterol, LDL cholesterol, HDL cholesterol and type 2 diabetes status. Exclusion criteria were active cardiac, liver, or renal disease or long-term complications from diabetes. None of the nondiabetic subjects were taking medications that may alter glucose tolerance. Diabetic subjects taking anti-diabetes medications (sulfonylurea or metformin) were asked to discontinue their medications for at least two weeks prior to participation.

The study was approved by the Human Investigation Ethics Committee of Huashan Hospital. With a full understanding of the study, each participant signed the informed consent form voluntarily.

**Isolation of white blood cells.** eBioscience™ 10X RBC Lysis Buffer (00-4300, Invitrogen, Carlsbad, USA) was used to isolate PBMCs according to the manufacture instruction. Briefly, 20 mL 1x RBC lysis buffer was mixed with 2 mL blood (with EDTA anticoagulant) and incubated 10 min at room temperature. After centrifuging at $500g$ at RT for 5 min, the pellets were resuspended with 1x RBC lysis buffer and centrifuged again. The pellets were then washed with PBS and finally used for ELISA.

Isolated WBCs were cultured in 6-well plates, using supplemented RPMI-1640 medium with 10% FBS, and incubated at 37 °C in a humidified, 5% $CO_2$ atmosphere.

## Assays

**ITT and GTT.** Assays were performed on male C57BL/6 mice after 12 weeks of treatments. For GTTs, mice were intraperitoneally injected with glucose (1 g/kg) after 16 h of fasting, and blood was sampled at 0, 15, 30, 60, 90, and 120 min after glucose injection. For ITTs, mice were intraperitoneally injected with 0.4–0.5 units/kg of insulin after 6 h of fasting, and blood was sampled at 0, 15, 30, 60, 90, and 120 min after insulin injection.

**Blood insulin and HOMA-IR.** Plasma insulin levels were measured by ELISA (Abcam, Cambridge, UK). The HOMA-IR was calculated as follows: Fasting blood glucose level (FPG, mmol/L) * Fasting insulin level (FINS, μU/mL) / 22.5.

**Glucose uptake.** The non-radioactive 2-dexoyglucose uptake was detected using a Glucose Uptake-Glo™ Assay Kit (J1341, Promega, Madison, USA) according to manufactured instruction. Briefly, cells were seed on 96-well plates and serum starved for 3 h before the assay. Cells were washed with PBS three times then replaced the medium with DMEM without serum or glucose and treatment with 2 mM methyl-phenylalanine 2 h, then stimulated with 100 nM insulin for 30 min. Cells were incubated with 1 mM 2-DG in PBS for additional 10 min at 25 °C. Cells were then lysed in stop buffer and neutralized with neutralization buffer. Lysates were then centrifuged at 15000 g for 15 min at 4 °C, and the supernatant were incubated with 2DG6P detection reagent for 1 h at 25 °C, then recorded luminescence with 0.3–1 s integration on a luminometer (Glomax96, Promega, Madison, USA).

**Myc-GLUT4-mCherry assay.** Cells were seeded on poly-D-lysine coated overslips were transfected with Myc-GLUT4-mCherry plasmid by electrotransfection. Cells were harvested and fixed with 4% paraformaldehyde for 10 min, then washed with cold PBS two times and blocked with PBS containing 5% donkey serum for 45 min at room temperature. Cells were then incubated with primary rabbit anti-Myc antibody overnight at 4 °C, followed by Alexa Fluor 488-conjugated Donkey anti-rabbit secondary antibody at least 1 h at room temperature. After washing, the overslips were mounted with DAKO mounting medium. Mounted samples were subjected to confocal imagining using a Zeiss LSM710 confocal laser microscope system with a x63 NA/1.40 CFI Plan APO VC oil-immersion objective.

**Immunofluorescence.** Cells were harvested and washed with PBS twice to remove the remaining medium. Paraformaldehyde (4%) was used to fix the cells at room temperature, 0.5% Triton X-100 in PBS was added and incubated for 20 min at room temperature. 3% BSA was added for 30 min at room temperature. The primary antibodies used were: anti-mouse IRβ (1:100, Abcam, Cambridge, UK) and anti-rabbit FARSA (1:100, Proteintech, Rosemont, USA). The secondary antibodies (1: 1000, Thermo Fisher Scientific) used were: Alexa Fluor 555 donkey anti-rabbit and Alexa Fluor 488 donkey anti-mouse. DAPI was subsequently added, for nuclear staining. Cells were observed under a fluorescence microscope.

**Immunostaining of GLUT4.** Mice muscle tissues were fixed in 4% paraformaldehyde in PBS for 4 h and in 30% sucrose overnight, embedded in opti-mum cutting temperature (O.C.T.) compound

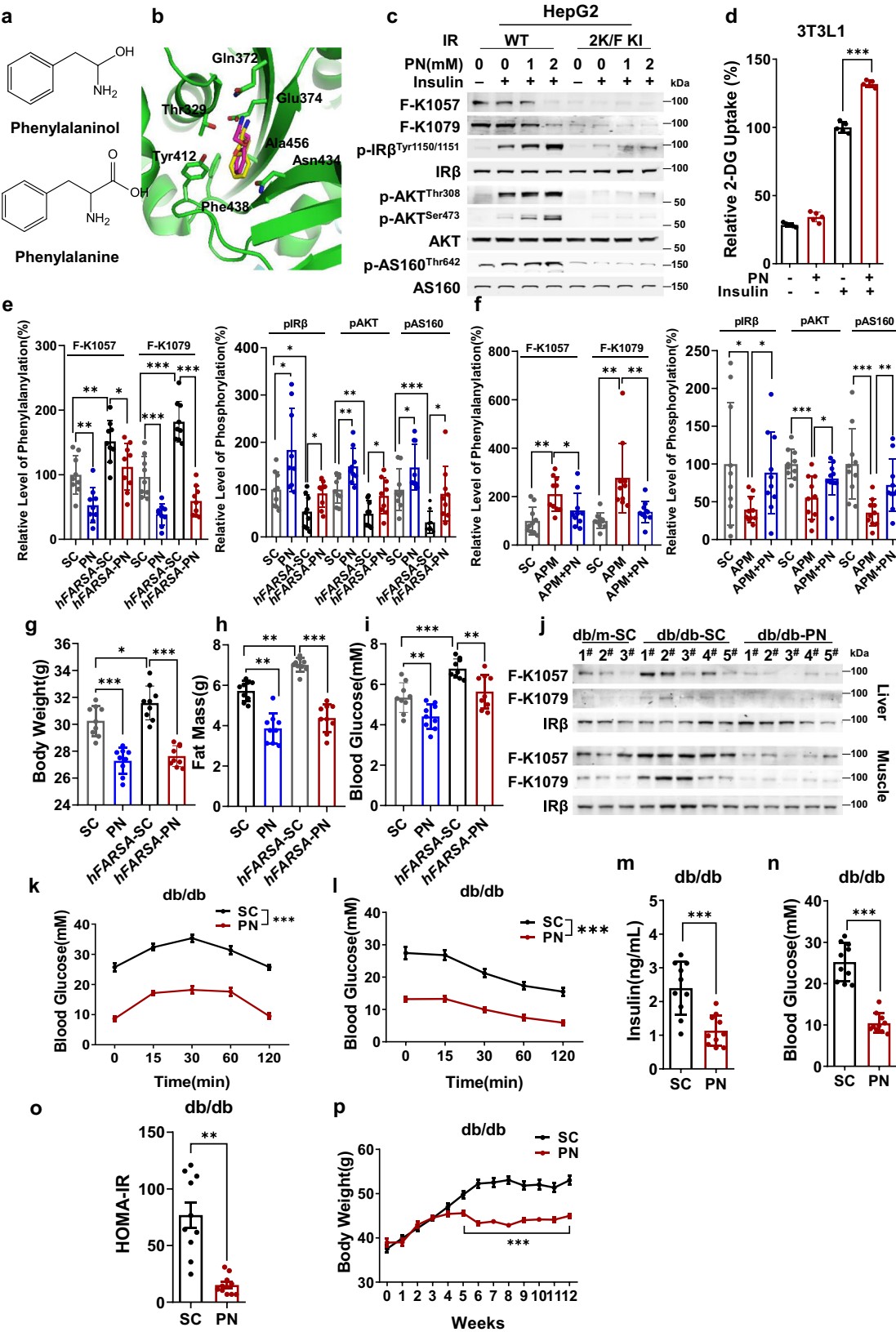

(Sakura) and cut into 5 μm thick sections. Sections were fixed in 4% PFA at RT for 30 min and permeabilized with 0.1% Triton X-100 in PBS for 10 min. After washing with PBS, sections were blocked in PBS containing 5% goat serum for 1 h at RT, and then incubated with the GLUT4 antibody (1:100 dilution in 5% goat serum) overnight at 4 °C. After washing with PBS, sections were incubated with Alexa Flour 488 goat anti-rabbit secondary antibody (Invitrogen, Carlsbad, USA) for 1 h at RT in dark. Sections were washed and then incubated with DAPI (P36962, Thermo Scientific, Carlsbad, USA) for 1 minute to display the nuclei. Sections were mounted in a fluorescent mounting medium and images were captured using a confocal laser scanning microscope (Olympus, FV3000).

**Fig. 7 | Decreasing F-K1057/1079 restores insulin signaling. a** Phenylalaninol is structurally analogous to phenylalanine. The molecular formulae of phenylalanine and phenylalaninol are shown. **b** Phenylalaninol and phenylalanine may occupy the same binding site of FARSA. Computer simulations show that phenylalaninol fits the phenylalanine binding site of FARSA. **c** Phenylalaninol decreases F-K1057/1079 levels and activates insulin signaling. The F-K1057/1079 levels of IRβ and phosphorylation levels of components of insulin signaling were detected in HepG2 cells and IRβ$^{2K/F}$ knock-in HepG2 cells in the presence of different levels of phenylalaninol. **d** Phenylalaninol activates glucose uptake. The 2-DG uptake of 3T3-L1 cells and insulin-stimulated 3T3-L1 cells was measured in the absence and presence of 2 mM phenylalaninol ($n = 5$). **e, f** Phenylalaninol decreases F-K1057/1079 levels and activates insulin signaling in h*FARSA*-transgenic and aspartame-fed mice. The liver F-K1057/1079 levels (left) and insulin signaling were quantified for male wild-type and h*FARSA*-transgenic mice that were fed with standard and phenylalaninol-

supplemented chow (**e**) and for male wild-type mice that were fed with standard, aspartame-supplemented, and chow supplemented with both aspartame and phenylalaninol (**f**). All values were normalized to those of wild-type mice fed standard chow ($n = 9$). **g–i** Phenylalaninol chow relieves diabetic symptoms in h*FARSA*-transgenic mice. Male wild-type and h*FARSA*-transgenic C57 mice were fed standard or phenylalaninol-supplemented chow for 12 weeks ($n = 9$). Body weight (**g**), fat mass (**h**), and fasting blood glucose level (**i**) were measured. **j–p** Phenylalaninol decreases F-K1057/1079 levels and relieves diabetic symptoms in db/db mice. Male db/db mice were fed standard or phenylalaninol-supplemented chow for 12 weeks ($n = 10$). The F-K1057/1079 levels in mouse liver and muscle tissues (**j**), glucose tolerance (**k**), insulin tolerance (**l**), blood insulin (**m**), blood glucose levels (**n**), HOMA-IR values (**o**), and body weights (**p**) were measured. Significance was indicated as two-tailed, unpaired, $t$ test for (**d**), (**k–p**), one-way ANOVA test for (**e–i**). Values are expressed as the mean ± SEM. *$p < 0.05$, **$p < 0.01$, ***$p < 0.001$.

**ELISA for F-K1057, F-K1079, SIRT1 and FARSA quantitation.** ELISA-based quantitation methods were developed for quantifying the F-K1057 and F-K1079 of PBMC in clinical samples. Microtiter-plate wells were coated with IRβ antibody (#3020, CST, Danvers, Massachusetts, USA). The PBMC lysate, F-K1057 or F-K1079 antibody and GR-HRP was loaded into the wells. After 2 h of incubation at 37 °C, the wells were washed and 3,30,5,50-tetramethylbenzidine (TMB) substrate solution was added into the wells. The F-K1057 and F-K1079 concentrations were obtained by comparing the absorbance of samples at 450 nm with that of standards.

For SIRT1 or FARSA quantification, we coated microtiter-plate wells with SIRT1 antibody (#60303-1-Ig, Proteintech, Rosemont, USA) or FARSA antibody (#sc-100987, Santacruz, Dallas, USA) and followed the same procedures as described for F-K1057 and F-K1079. SIRT1(#13161-1-AP, Proteintech, Rosemont, USA) or FARSA (#18121-1-AP, Proteintech, Rosemont, USA) was used as the standards for each assay, respectively.

**Immunoprecipitation.** Cells or tissues were homogenized in lysis buffer cells were lysed with 0.5% NP-40 buffer (50 mM Tris-HCl (pH 7.5), 150 mM NaCl, 0.5% NP-40, 1 μg/mL aprotinin, 1 μg/mL leupeptin, 1 μg/mL pepstatin, and 1 mM PMSF. Protein was incubated with anti-IRS1 antibody. The binding complexes were washed with 0.5% NP-40 buffer and mixed with loading buffer for SDS-PAGE. Presence of IRS1 in the precipitates was detected by a rabbit anti-IRS1 and anti-pan-phospho-tyrosine antibody.

**Western blot.** Cultured cells or cells extracted from mice tissue and patient-matched PBMC were homogenized with 0.5% NP-40 buffer containing 50 mM Tris-HCl (pH 7.5), 150 mM NaCl, 0.5% Nonidet P-40, and a mixture of protease inhibitors (Sigma-Aldrich). After centrifugation at 12,000 rpm and 4 °C for 15 min, the supernatant of the lysates was collected for western blotting according to standard procedures. Detection was performed by measuring chemiluminescence on a Typhoon FLA 9500 (GE Healthcare, Little Chalfont, UK).

**In Vitro Phenylalanylation Reaction.** In vitro phenylalanylation reactions were carried out in a 30 μl reaction mix that contained 50 mM HEPES (pH 7.5), 25 mM KCl, 2 mM MgCl₂, 5 mM phenylalanine, 4 mM ATP, 10 nM FARSA/B, 0.05 mg/ml synthetic substrate peptide or recombinant IR protein purified from HEK293T cells as final concentrations. The pH of the reaction mix was adjusted to 7.5 before starting the reaction through adding FARSA/B. The reaction was allowed continue for 3 h at 37 °C. The peptide was desalted by passing through a C18 ZipTip (Millipore) before subjecting to analysis by a MALDI-TOF/TOF mass spectrometer (SCIEeX-5800) and the resulted IR was subjected to western blot analysis.

**In vitro de-phenylalanylation reaction.** In vitro de-phenylalanylation reactions were carried out in a 30 μl reaction mix that contained

50 mM HEPES (pH 7.5), 6 mM MgCl₂, 1 mM DTT, 1 mM NAD⁺, 0.05 mg/ml synthetic phenylalanylated peptide or recombinant IR protein, 1 mg/ml SIRT1 and 1 mM PMSF. The reaction was allowed continue for 4 h at 37 °C. The peptide was desalted by passing through a C18 ZipTip (Millipore) before subjecting to analysis by a MALDI-TOF/TOF mass spectrometer (SCIEeX-5800) and the resulted IR was subjected to western blot analysis.

**Nuclear magnetic resonance analysis of sera metabolites.** Nuclear magnetic resonance (NMR) analysis was adapted from published methods[51](4). Briefly, Metabolites in 200 μl sera samples were extracted by adding 800 μL prechilled methanol. Supernatants were collected after centrifugation (12,000$g$, 4 °C) for 10 min. The extraction was repeated twice and the supernatants were combined and lyophilized after vacuum dry to remove methanol. The extracted metabolites were redissolved in 600 μl phosphate buffer (0.15 M, pH 7.4) by vortex. The resulted solution was centrifuged at 16,099$g$ (4 °C) for 10 min before 550 μl supernatant was transferred into NMR tubes for analysis. All the one-dimensional 1H NMR spectra were acquired at 298 K on a Bruker Advance III 600 MHz NMR spectrometer (600.13 MHz for proton frequency) equipped with an inverse cryogenic probe (Bruker Biospin, Germany) using the first increment of the gradient selected NOESY pulse sequence (NOESYGPPR1D: recycle delay-G1-900-T1-900-tm-G2-900-acquisition). 64 transients were collected into 32 k data points with a spectral width of 20 ppm for each sample. All NMR spectra were processed using the spectral width of 20 ppm for each sample. All NMR spectra were processed using the software package TOPSPIN (V3.6.0 software package TOP-SPIN (V3.6.0) (Bruker Biospin, Bruker Biospin, Karlsruhe, Karlsruhe, Germany). For 1H NMR spectra, an exponential window function was employed with a line broadening factor of 1 Hz and zeroHz and zero-filled to 128 K prior to Fourier transformation. Each spectrum was then filled to 128 K prior to Fourier transformation. The characteristic and least-overlapping NMR signals (tyrosine: δ 6.89; phenylalanine: δ 7.43) were used to calculate absolute concentration of metabolites with the known concentration of TSP. For NMR signals tyrosine and phenylalanine were used to study the concentration of metabolites wtih the known concentration of TSP, software package TOPSPIN soft.

**MS/MS Identification of F-K1057 and F-K1079**
**Sample preparation.** Purified FARS were digested with trypsin, followed by LC-MS/MS analysis. MaxQuant (version 1.4.1.2) was employed to search MS/MS spectra against the SwissProt-human database (Release 2014-04-10) that contained forward and reverse sequences and oxidation of methionine, using the Andromeda search engine. Mass and fragment mass had an initial mass tolerance of 5 ppm and 0.05 Da. Minimal peptide length was set to seven amino acids and a maximum of four mis-cleavages was allowed. The false discovery rate (FDR) was set to 0.01 for peptide and protein identifications.

IR was ectopically expressed in HepG2 cells that were cultured in media without or with 2 mM methyl-phenylalanine. Cells were harvested and lysed in 0.1% NP-40 buffer (50 mM Tris-HCl, pH=7.5, 150 mM NaCl, 0.1% Nonidet P 40) and anti-FLAG M2 magnetic beads (M8823-1ml, Sigma-Aldrich, Darmstadt, Germany) was employed to precipitate IR proteins for 3 h at 4 °C. The beads were then washed twice with 0.1% NP-40 buffer, twice with ddH$_2$O and three times with 50 mM NH$_4$HCO$_3$, followed by on-bead trypsin digestion at 37 °C overnight. The resulted peptides in the supernatants were collected through centrifugation and dried in a speed vacuum (Eppendorf, Hamburg, Germany). Obtained peptides were stored at −80 °C for LC-MS/MS analysis.

**LC-MS/MS analysis.** LC peaks eluted at the same time as those of standard synthetic peptides were subject to MS analysis with Orbitrap Fusion mass spectrometer. Peptide counting and/or areas of samples were compared to those of standard synthetic peptides of known concentrations. The MS machine was operated in the data-dependent mode to switch automatically between MS and MS/MS acquisition. Survey full-scan MS spectra ($m/z$ 350–1600) were acquired in Orbitrap with a mass resolution of 60, 000 at $m/z$ 200. The AGC target was set to 300, 000, and the maximum injection time was 50 ms. MS/MS acquisition was performed in Orbitrap with 3 s cycle time, the resolution was 15, 000 at $m/z$ 200. The intensity threshold was 50,000, and the maximum injection time was 200 ms. The AGC target was set to 200,000, and the isolation window was 2 $m/z$. Ions with charge states 2+, 3+, and 4+ were sequentially fragmented by higher energy collisional dissociation with a normalized collision energy of 30%, fixed first mass was set at 120. In all cases, one micro scan was recorded using dynamic exclusion of 30 s.

Raw mass spectrometry files were searched using Protein Discoverer(version 1.4.0.288, Thermo Fisher Scientific) with Mascot (version 2.7.0, Matrix Science). The data were processed with UniProt Human protein database (70727 entries, download in 20161202). The mass tolerances were 10 ppm for precursor and fragment Mass Tolerance 0.05 Da. Up to two missed cleavages were allowed. The search engine set protein N-acetylation, pyroglutamate on peptide N termini and oxidation on methionine and phenylalanylation of lysine as variable modifications, the cysteine carbamidomethylation as a fixed modification. The mass spectrometry proteomics data have been deposited to the ProteomeXchange Consortium via the PRIDE partner repository[52] with the dataset identifier PXD035076.

Quantification of targeted phenylalanylated peptide was achieved by employing a published method[53]. Briefly, a ratio of phenylalanylated peptide signal (the total ion counts (TIC) of phenylalanylated form) to the total peptide signal (TIC of phenylalanylated form + TIC of non-phenylalanylated form) were calculated according to the following equation: TICK-Phe/ (TICK-Phe + TICnon-K-Phe) = Ratio of K-Phe (RK-Phe).

**Structural modeling.** Due to the lack of structural information of FARSA, the binding of phenylalaninol to the phenylalanine binding site of FARSA was modeled with Protein Homology/analogY Recognition Engine V 2.0 (Phyre2).

**Statistics and reproducibility**
Statistical analysis was performed using Prism 8.0 (GraphPad Software, Inc., San Diego, CA, USA.) and Excel (Microsoft Corp., Redmond, CA, USA). Two-tailed Student's $t$ test was performed for the two-group analysis (Figs. 1a-g, 2b, 2d, 2g, 2h, 2i-k, 3k, 6a, 6e, 6g, 6i, 6k, 7d, 7k-p and Supplementary Figs. 1b-e, 1g-m, 1o-t, 2a-c, 2f, 3j-k, 7d, 8c-g, 9b and Table S1). One-way Welch's ANOVA to compare more than two groups (Figs. 2f, 3b-c, 4d, 4f, 4h, 5i, 7e-i and Supplementary Fig. 3f–i). Pooled results were expressed as the means ± SEM. Differences were considered statistically significant if the $p$ value was less than 0.05. Significance was indicated as *$p < 0.05$; **$p < 0.01$; ***$p < 0.001$.

Western blot assays were performed at least two independent times. Five or more regions of interest were randomly assessed for each microscopy image, and representative images were shown. To enhance reproducibility, no less than five animals were assigned per group.

**Reporting summary**
Further information on research design is available in the Nature Research Reporting Summary linked to this article.

## Data availability
The mass spectrometry proteomics data have been deposited to the ProteomeXchange Consortium via the PRIDE partner repository with the dataset identifier PXD035076. All experimental data that support the findings of this study are included as follows: 7 main figures, 10 Supplementary figures, and Source data. Source data are provided with this paper.

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

## Acknowledgements

This work was supported by Grants from the State Key Development Programs of China (Nos. 2018YFA0801300 (S.M.Z.), 2018YFA0800300 (W.X.)), the National Science Foundation of China (Nos.31821002 (S.M.Z.), 92157001 (S.M.Z.), 91857000 (S.M.Z.), 31930062 (S.M.Z.), 31871432 (W.X.), 32171298 (W.X.), 81971449 (Y.Y.Y.), 82171672 (Y.Y.Y.)), Program of Shanghai Academic Research Leader(21XD1423000 (W.X.)), Innovation-oriented Science and Technology Grant from Key Laboratory of Reproduction Regulation of NHC (CX2017-0X) and grant from Key Laboratory of Reproduction Regulation of NPFPC(S.M.Z).

## Author contributions

S.M.Z. and W.X. conceived the project and designed the experiments. S.M.Z., W.X., and Q.Z. supervised experiments, analyzed the data and wrote the manuscript. Q.Z. executed molecular biological experiment and analysis and interpretation of data. Q.Z., J.C.C., P.C.L., W.X.S.

supervised animal experiments. W.W.S., Q.Z., J.C.C., H.L.Z., J.L., Y.L. supervised clinical sample experiments. Q.Z., Y.P.A., L.H., and X.W.Z. performed the NMR, LC-MS/MS, and MALDI-TOF/TOF mass analysis. B.X.W. conducted the computer modeling. Y.M.L., Y.Y.Y., and J.Y.Z. discussed and contributed to the interpretation of the data. All the authors discussed the results and commented on the manuscript.

## Competing interests

The authors declare no competing interests.
