## [Peer Review File · Nature Communications]

Phenylalanine Impairs Insulin Signaling and Inhibits Glucose Uptake Through Modification of IR βEditorial Note: This manuscript has been previously reviewed at another journal that is not operating a transparent peer review scheme. The manuscript was considered suitable for publication without further review at *Nature Communications*.

REVIEWER COMMENTS

Reviewer #2 (Remarks to the Author):

Major

1. The representation of the dose of Phenylalanine and Aspartame persist, though some of the experiments do address this. Lines 364-367 do not clarify the limitations of the experimental model (unclear to me if the line reference is incorrect or if the changes were not made) as stated in the responses to points 3 and 5. Nonetheless, the data in extended figure 1e-k do support the impact of chronic Phe overfeeding in a more physiologically relevant range (~ 20-25% increase). The statement of “8-101% phenylalanine elevation in T2D” (line 85) could be modified to more accurately state that this range represents the extremes for serum Phe elevation in human samples.

2. The writing is much clearer and the Discussion more balanced than prior iterations. I think the authors still miss an opportunity to discuss some of the complexities of their studies. The data they generate suggest that phenylation is yet another covalent modification that may impact protein function and joins the list of other better known modifications, including phosphorylation, sumo-lytation, myristoylation, OGlcNAc-ylation, ubiquitination, etc. Like OGlcNAcylation, phenylation may signal excess nutrient supply and serve to shift metabolic signals away from energy storage. The data from extended Figure 7e still suggest to me a disassociation between F-K of the insulin receptor and the canonical downstream targets (i.e HF+PN seems to block F-K but does not restore pIRb, pAkt or pAS160 signals). Other pathways may be still interfering with insulin signaling with a high-fat diet. So, K-F is one of many signals that helps form an integrated response to nutrient excess.

It would be interesting to also consider some of the hints the authors see with regards to energy balance. Though Phe supplementation did not seem to induce weight gain normal C57BL/6 mice (extended figure 1b and e), Aspartame did (Extended Figure 1m) and perhaps related to increased food intake (extended Figure 1n). The data from extended figure 3e-i suggest that overexpressing hFARSA tg mice seems to promote weight gain (though paradoxically, also energy expenditure) whereas PN decreases weight gain with a decrease in EE. Neither of these perturbations seemed to impact food intake or RQ. Similar PN supplementation decrease weight in db/db mice (Figure 7p). Thus, there appears to be a relationship (albeit imperfect) between F-K and energy balance.

This will not be trivial to sort out. For example, the data the authors have suggest the increase in weight gain with increased EE (extended figure 3i).. The increases in EE are likely greater than actually reported since the value is normalized for a higher body weight and thus numerically lower. If the authors were to plot EE per mouse against body weight for each group (see Tschop MH et al., Nat Method 2011, PMID: 22205519, also <https://www.mmpc.org/shared/regression.aspx>) . While a conclusive explanation for the impact on energy balance (and perhaps energy efficiency) is beyond the scope of this study, a brief acknowledgement that K-F may impact energy balance would be helpful in the discussion.

Minor

1. The statement that “glucose is the preferred energy source” (Ln 22-23) is misleading. Substrate utilization depends on a multitude of factors.. cell type, fasting/fed states, substrate availability, etc. This sentence should just be omitted.
2. While I agree that odd-chain fatty acids can enter gluconeogenesis via propionate, the sentence “Fatty acids and amino acids are two other major energy source that can be converted to glucose...” (Ln 38-39) does not clearly convey this (despite the assurances the response letter). Odd chain fatty acids only represent a very small portion of dietary fat or perhaps the products of gut microbial metabolism. Why not simply replace “fatty acids” with “Glycerol, from triglyceride”. That
3. Ln 64: “Glucose transportation” should be “glucose transport”.
4. Ln 67: “Fatty acids alter cell signaling by modifying proteins” is misleading. Fatty acids can do more than this. Fatty acids can activate myriad pathways that alter cell function.

Point-by-Point responses to REVIEWER COMMENTS

Reviewer #2 (Remarks to the Author):

Major

1. The representation of the dose of Phenylalanine and Aspartame persist, though some of the experiments do address this. Lines 364-367 do not clarify the limitations of the experimental model (unclear to me if the line reference is incorrect or if the changes were not made) as stated in the responses to points 3 and 5. Nonetheless, the data in extended figure 1e-k do support the impact of chronic Phe overfeeding in a more physiologically relevant range (~ 20-25% increase). The statement of “8-101% phenylalanine elevation in T2D” (line 85) could be modified to more accurately state that this range represents the extremes for serum Phe elevation in human samples.

Response:

Following the reviewer’s suggestion, we add the following paragraph to the Discussion session: “The limitations of the current study include that 1% in dietary exposure of Phe or aspartame to induce type 2 diabetes phenotypes is beyond real life exposures, and this may reduce the reliability of the conclusion. Second, we noticed that F-K1057/1079 slightly affect energy balance, as evidenced by altered weight gain and energy expenditure was found in overexpressing hFARSA tg and PN-fed mice (Supplementary Fig 3e-i), which leaves a possibility that altered energy balance may also contribute to F-K1057/1079-mediated insulin sensitivity.”

Moreover, following the reviewer’s suggestion, we modified the sentence of revised line 83 to “simulating 20-25% phenylalanine elevation in T2D human sera samples”.

2. The writing is much clearer and the Discussion more balanced than prior iterations.

I think the authors still miss an opportunity to discuss some of the complexities of

their studies. The data they generate suggest that phenylation is yet another covalent modification that may impact protein function and joins the list of other better known modifications, including phosphorylation, sumo-lytation, myristoylation, OGlcNAc-ylation, ubiquitination, etc. Like OGlcNAcylation, phenylation may signal excess nutrient supply and serve to shift metabolic signals away from energy storage. The data from extended Figure 7e still suggest to me a disassociation between F-K of the insulin receptor and the canonical downstream targets (i.e HF+PN seems to block F-K but does not restore pIRb, pAkt or pAS160 signals). Other pathways may be still interfering with insulin signaling with a high-fat diet. So, K-F is one of many signals that helps form an integrated response to nutrient excess.

Response:

Following the reviewer's suggestion, we added one sentence to make it clear that Phenylalanylation is one of many signals that helps form an integrated response to nutrient excess. "For IR β , the phenylalanine signal F-K1057/1079 inactivates insulin signaling and inhibits glucose uptake, providing a protective mechanism of excess glucose uptake when intracellular amino acids are abundant. Therefore, phenylalanylation, together with already reported mechanisms including phosphorylation, sumolytation, myristylation, OGlcNAc-ylation, ubiquitination of IR β , provided another integrated response to nutrient excess."

3, It would be interesting to also consider some of the hints the authors see with regards to energy balance. Though Phe supplementation did not seem to induce weight gain normal C57BL/6 mice (extended figure 1b and e), Aspartame did (Extended Figure 1m) and perhaps related to increased food intake (extended Figure 1n). The data from extended figure 3e-I suggest that overexpressing hFARSA tg mice seems to promote weight gain (though paradoxically, also energy expenditure) whereas PN decreases weight gain with a decrease in EE. Neither of these

perturbations seemed to impact food intake or RQ. Similarity PN supplementation decrease weight in db/db mice (Figure 7p). Thus, there appears to be a relationship (albeit imperfect) between F-K and energy balance.

This will not be trivial to sort out. For example, the data the authors have suggest the increase in weight gain with increased EE (extended figure 3i).. The increases in EE are likely greater than actually reported since the value is normalized for a higher body weight and thus numerically lower. If the authors were to plot EE per mouse against body weight for each group (see Tschop MH et al., Nat Method 2011, PMID: 22205519, also <https://www.mmpc.org/shared/regression.aspx>) . While a conclusive explanation for the impact on energy balance (and perhaps energy efficiency) is beyond the scope of this study, a brief acknowledgement that K-F may impact energy balance would be helpful in the discussion.

Response:

We agree with the reviewer that F-K may have impact on energy balance, so we add the following paragraph to the revised discussion. It reads as “The limitations of the current study include that 1% in dietary exposure of Phe or aspartame to induce type 2 diabetes phenotypes is beyond real life exposures, and this may reduce the reliability of the conclusion. Second, we noticed that F-K1057/1079 slightly affect energy balance, as evidenced by altered weight gain and energy expenditure was found in overexpressing hFARSA tg and PN-fed mice (Supplementary Fig 3e-i), which leaves a possibility that altered energy balance may also contribute to F-K1057/1079-mediated insulin sensitivity.”

Following the reviewer’s suggestion, we measured 24 h real-time EE of mice at room temperature (the graphs is shown below). ANCOVA revealed that mice feeding with PN diet had significantly lower total EE estimated in the common lean mass than SC diet. Meanwhile, hFARSA transgenic mice had higher EE. These results suggest that K-Phe may increase EE to impact energy balance. However, it will be premature to draw a definitive conclusion based on the current results and that is somehow beyond

the focal point of the current study, we show the graphs to the reviewer for his/her information.

Minor

1. The statement that “glucose is the preferred energy source” (Ln 22-23) is misleading. Substrate utilization depends on a multitude of factors.. cell type, fasting/fed states, substrate availability, etc. This sentence should just be omitted.

Response: This sentence had been omitted.

2. While I agree that odd-chain fatty acids can enter gluconeogenesis via propionate, the sentence “Fatty acids and amino acids are two other major energy source that can be converted to glucose...” (Ln 38-39) does not clearly convey this (despite the assurances the response letter). Odd chain fatty acids only represent a very small portion of dietary fat or perhaps the products of gut microbial metabolism. Why not simply replace “fatty acids” with “Glycerol, from triglyceride”.

Response: Modification had been made following the reviewer's suggestion.

3. Ln 64: "Glucose transportation" should be "glucose transport".

Response: We have corrected it.

4. Ln 67: "Fatty acids alter cell signaling by modifying proteins" is misleading. Fatty acids can do more than this. Fatty acids can activate myriad pathways that alter cell function.

Response: We modified this sentence to "Fatty acids can modify proteins as substrates to alter cell signaling".

REVIEWERS' COMMENTS

Reviewer #2 (Remarks to the Author):

All concerns have been addressed.